# K63-linked ubiquitination regulates RIPK1 kinase activity to prevent cell death during embryogenesis and inflammation

Yong Tang[1,3], Hailin Tu[1,3], Jie Zhang[1], Xueqiang Zhao[1], Yini Wang[2], Jun Qin[2] & Xin Lin [1]

Receptor-interacting protein kinase 1 (RIPK1) is a critical regulator of cell death through its kinase activity. However, how its kinase activity is regulated remains poorly understood. Here, we generate $Ripk1^{K376R/K376R}$ knock-in mice in which the Lys(K)63-linked ubiquitination of RIPK1 is impaired. The knock-in mice display an early embryonic lethality due to massive cell death that is resulted from reduced TAK1-mediated suppression on RIPK1 kinase activity and forming more TNFR1 complex II in $Ripk1^{K376R/K376R}$ cells in response to TNFα. Although TNFR1 deficiency delays the lethality, concomitant deletion of RIPK3 and Caspase8 fully prevents embryonic lethality of $Ripk1^{K376R/K376R}$ mice. Notably, $Ripk1^{K376R/-}$ mice are viable but develop severe systemic inflammation that is mainly driven by RIPK3-dependent signaling pathway, indicating that K63-linked ubiquitination on Lys376 residue of RIPK1 also contributes to inflammation process. Together, our study reveals the mechanism by which K63-linked ubiquitination on K376 regulates RIPK1 kinase activity to control cell death programs.

---

[1] Institute for Immunology, Tsinghua University School of Medicine, Tsinghua University-Peking University Jointed Center for Life Sciences, 100084 Beijing, China. [2] State Key Laboratory of Proteomics, Beijing Proteome Research Center; National Center for Protein Sciences (The PHOENIX Center, Beijing), Institute of Lifeomics, 102206 Beijing, China. [3] These authors contributed equally: Yong Tang, Hailin Tu. Correspondence and requests for materials should be addressed to X.L. (email: linxin307@tsinghua.edu.cn)

Receptor-interacting protein kinase 1 (RIPK1) is a serine/threonine kinase that plays a critical role in various immune signaling pathways to regulate cell survival or cell death[1]. Its function has been extensively studied in the tumor necrosis factor receptor 1 (TNFR1) signaling pathway[2]. The activation of TNFR1 signaling pathway can regulate two opposite downstream events, including forming the TNFR1 complex I (Complex I) that mediates nuclear factor-κB (NF-κB) and mitogen-activated protein kinase (MAPK) pathway activation for cell survival[3–5], and forming the TNFR1 complex II (Complex II) that mediates apoptosis or necroptosis for cell death[6–9]. RIPK1 kinase activity is critical for regulating the transition from complex I to Complex II. However, the regulation of RIPK1 kinase activity remains poorly understood.

Previous studies reported that phosphorylation of RIPK1 can regulate RIPK1 kinase activity. IκB kinase α/β (IKKα/β) complex directly phosphorylates RIPK1 in Complex I and protects cells from RIPK1 kinase-dependent death. Inhibition of IKKα/β could enhance RIPK1 kinase activity to promote Complex II formation[10]. Another kinase MK2 could directly phosphorylate RIPK1 at S321 and S336 to repress RIPK1 kinase-dependent death, and phosphorylation of RIPK1 by MK2 restricted its cytosolic activation and subsequent integration into Complex II by inhibiting RIPK1 kinase activity[11–13]. A recent finding reported that TBK1 (TANK binding kinase 1) and IKKε phosphorylate RIPK1 to prevent RIPK1-dependent cell death[14]. Thus, phosphorylation of RIPK1 is responsible for its kinase activity and function transition.

The ubiquitination status of RIPK1 also has been implicated to regulate its function transition. Several types of ubiquitination have been shown in RIPK1, including K11-, K48-, K63-, and Met1-linked ubiquitination[15]. Cellular inhibitor of apoptosis protein-1 (cIAP1) and cIAP2 have been reported to be the direct E3 ligases for RIPK1 to modify K63-linked ubiquitination[16–18]. K63-linked ubiquitination of RIPK1 is required for tumor necrosis factor-α (TNFα)-induced NF-κB activation, and also inhibits cell death[19–21]. Linear ubiquitin chain assembly complex (LUBAC) can conjugate Met1-linked ubiquitination chains to RIPK1 to promote NF-κB activation[22,23]. Deficiency of LUBAC components could enhance RIPK1 kinase activity and promote Complex II formation[24,25]. Other E3 ubiquitin ligases including mind bomb-2 (MIB2) and HECT domain and ankyrin repeat-containing E3 ligase 1 (HACE1) also implicate in RIPK1-mediated cell death[26,27]. Besides, the deubiquitinases (DUBs) are involved in the regulation of RIPK1 activation. For example, A20 can mediate the cleavage of K63-linked ubiquitination chains on RIPK1[28], and CYLD can cleave both K63 and Met1 ubiquitination chains on RIPK1[29,30]. A recent study reported that ABIN-1 regulated RIPK1 activation by linking Met1-linked ubiquitination with K63-linked deubiquitination, suggesting a crosstalk between different ubiquitination chains on the regulation of RIPK1 activation[31].

We and others have previously reported that Lys377 (K377) in human RIPK1, which is conserved as K376 in murine RIPK1, is the residue for K63-linked ubiquitination in RIPK1[19,20]. In the present study, we determine the physiological function of K63-linked ubiquitination on RIPK1 by generating $Ripk1^{K376R/K376R}$ knock-in mice. These knock-in mice display early embryonic lethality due to severe cell death. We elucidate the molecular mechanism that K63-linked ubiquitination on K376 of RIPK1 can promote tumor growth factor-β-activated kinase 1 (TAK1)-mediated suppression on RIPK1 kinase activity. Our study reveals the physiological function and molecular mechanism of K63-linked ubiquitination of RIPK1 on K376 during embryogenesis and systemic inflammation.

## Results

**$Ripk1^{K376R/K376R}$ mice are embryonically lethal.** To determine the physiological function of K63-linked ubiquitination on RIPK1, we generated $Ripk1^{K376R/K376R}$ knock-in mice by using CRISPR-Cas9 technique that also resulted in a strain of RIPK1-knockout (KO) mice, in which the translation of RIPK1 stopped at D367 in the intermediate domain (Fig. 1a). It has been shown that RIPK1-KO mice were perinatally lethal due to the cell death in multiple organs and severe inflammation[32,33]. Unexpectedly, $Ripk1^{K376R/K376R}$ mice could not be detected when they were 1 month old (Fig. 1b), which suggested that $Ripk1^{K376R/K376R}$ mice might be embryonically lethal. To determine the exact embryonic stage at which $Ripk1^{K376R/K376R}$ mice die, we analyzed the embryos from different days of gestation. No significant morphological differences could be detected at E9.5–E11.5 between $Ripk1^{K376R/K376R}$ and $Ripk1^{+/+}$ embryos (Fig. 1c). However, from E12.5, $Ripk1^{K376R/K376R}$ embryos became smaller and had aberrant morphology compared to $Ripk1^{+/+}$ embryos (Fig. 1c). The number of $Ripk1^{K376R/K376R}$ embryos were significantly decreased at E13.5 and no $Ripk1^{K376R/K376R}$ embryos can be observed at E14.5 (Supplementary Fig. 1). Hematoxylin and eosin (H&E) staining results showed that the abnormalities of $Ripk1^{K376R/K376R}$ embryos were observed mainly in the liver region at E12.5 (Fig. 1d). Further analysis by terminal deoxynucleotidyl transferase dUTP nick-end labeling (TUNEL) staining showed that $Ripk1^{K376R/K376R}$ embryos had more TUNEL-positive cells than $Ripk1^{+/+}$ embryos, especially in liver section, suggesting massive cell death in liver (Fig. 1e). Consistently, $Ripk1^{K376R/K376R}$ embryos also had more cleaved Caspase8-positive regions (Fig. 1f). In addition, the inflammatory cytokines and chemokines, including chemokine (C-X-C motif) ligand 1 (CXCL1), CXCL2, interleukin-6 (IL-6), TNFα, interferon-β (IFNβ), and IFNγ were significantly increased in $Ripk1^{K376R/K376R}$ embryos (Fig. 1g). Collectively, these data suggest that $Ripk1^{K376R/K376R}$ mice died around E13.5 resulted from excessive cell death and severe inflammation.

**$Ripk1^{K376R/K376R}$ mutation promotes apoptosis and necroptosis.** To further investigate the exact molecular mechanism, we generated immortalized mouse embryonic fibroblast (MEF) cells derived from littermates of E11.5 wild-type (WT) and $Ripk1^{K376R/K376R}$ embryos. We also generated $Ripk1^{-/-}$ MEFs from E11.5 embryos. Although a truncated fragment of RIPK1 was expressed in $Ripk1^{-/-}$ MEFs (Supplementary Fig. 2a), it cannot interact with RIPK3 (Supplementary Fig. 2b), indicating null function of this truncated RIPK1. Since $Ripk1^{K376R/K376R}$ embryos died around E13.5 due to massive cell death, we postulated that $Ripk1^{K376R/K376R}$ MEFs were more sensitive to TNFα-induced cell death. With the stimulation by TNFα alone, $Ripk1^{K376R/K376R}$ MEFs showed higher levels of cell death and Caspase3 activity, even higher than $Ripk1^{-/-}$ MEFs (Fig. 2a and Supplementary Fig. 2c). Furthermore, TNFα plus cycloheximide (CHX) sensitized $Ripk1^{K376R/K376R}$ MEFs to cell death, but caspase inhibitor zVAD.fmk did not inhibit it (Fig. 2b). Treatment with TNFα/CHX/zVAD can induce necroptosis independently of caspases, and RIPK1 kinase inhibitor Nec-1 fully blocked the increased cell death in $Ripk1^{K376R/K376R}$ MEFs (Fig. 2b), suggesting that K376R mutation in RIPK1 sensitized cells to necroptosis mediated by RIPK1 kinase activity. Moreover, with TNFα/CHX stimulation, $Ripk1^{K376R/K376R}$ cells had more Caspase3 activity, which indicated more apoptosis (Fig. 2c).

We next examined the involvement of cell death machinery that contributes to TNFα-induced cell death in $Ripk1^{K376R/K376R}$ MEFs. Similar to RIPK1-deficient MEFs, $Ripk1^{K376R/K376R}$ MEFs produced more cleaved Caspase3 than WT control after TNFα stimulation (Fig. 2d), suggesting that $Ripk1^{K376R/K376R}$ mutation could

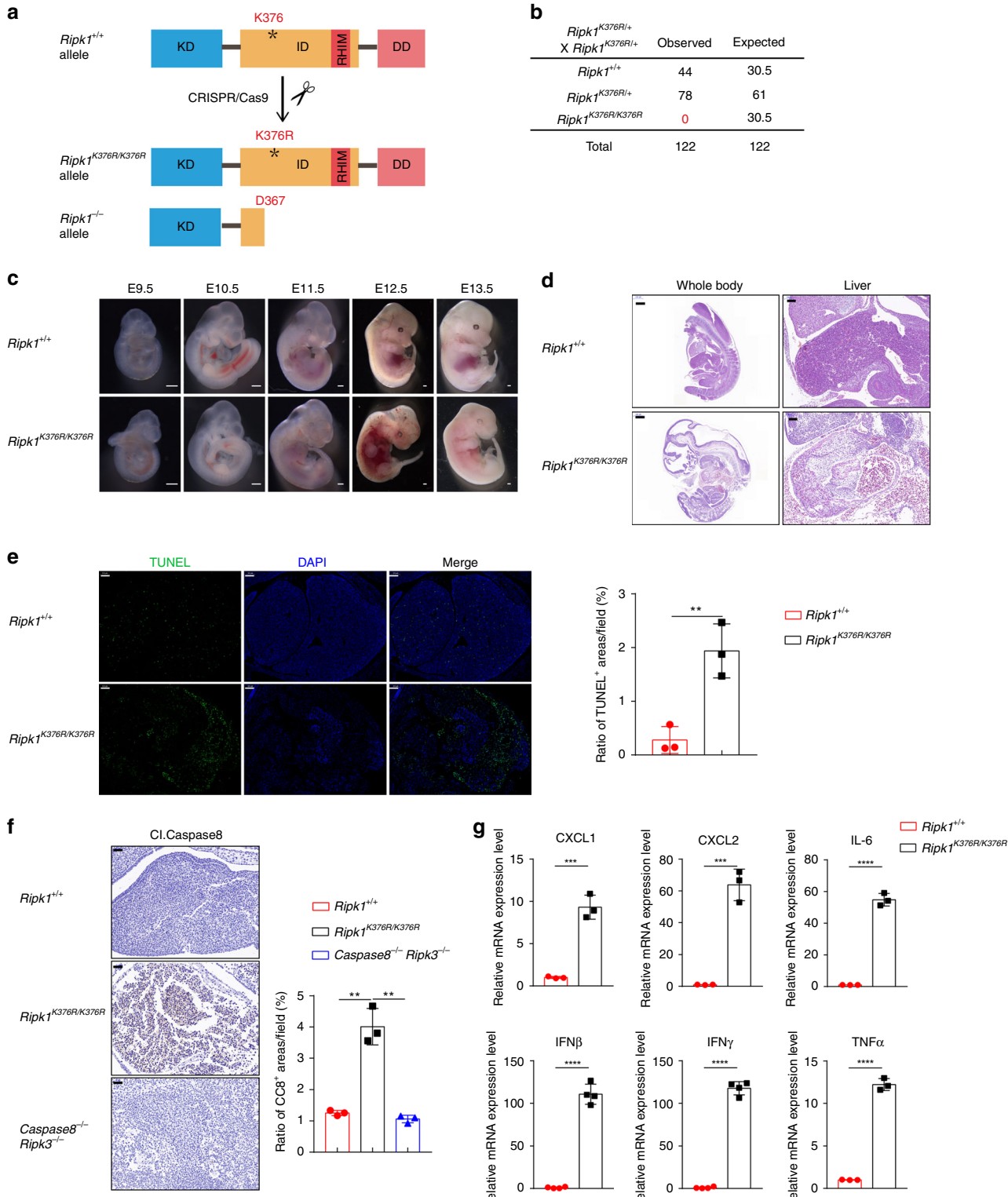

**Fig. 1** *Ripk1^{K376R/K376R}* mutation results in embryonic lethality. **a** Schematic overview of strategy to generate *Ripk1^{K376R/K376R}* and *Ripk1^{−/−}* mice by CRISPR-Cas9 technology. KD, kinase domain; ID, intermediate domain; DD, death domain; RHIM, RIP homotypic interaction motif. **b** Statistical analysis of the expected and observed offspring mice (1-month-old) from the intercrosses of *Ripk1^{K376R/+}* mice. **c** The representative images of embryos with the indicated genotypes from E9.5 to E13.5 (scale bar, 1 μm). **d** Hematoxylin and eosin (H&E) staining of embryos (left, scale bar, 500 μm) and liver sections (right, scale bar, 100 μm) at E12.5. **e** Microscopic images and statistical results of TUNEL staining in liver sections at E12.5 (scale bar, 100 μm; *Ripk1^{+/+}* embryo, n = 3; *Ripk1^{K376R/K376R}* embryo, n = 3). **f** Microscopic images and statistical results of three different area of cleaved Caspase8 staining of embryos at E12.5 (scale bar, 50 μm; *Ripk1^{+/+}* embryo: n = 3; *Ripk1^{K376R/K376R}* embryo: n = 3; *Ripk3^{−/−}Caspase8^{−/−}* embryo: n = 3). **g** qRT-PCR analysis of inflammatory cytokines and chemokines expression in embryo homogenates at E12.5. Data are mean ± s.e.m. (*Ripk1^{+/+}* embryo: n = 3; *Ripk1^{K376R/K376R}* embryo: n = 3). Statistical significance was determined using a two-tailed unpaired *t* test, **P < 0.01; ***P < 0.001, ****P < 0.0001

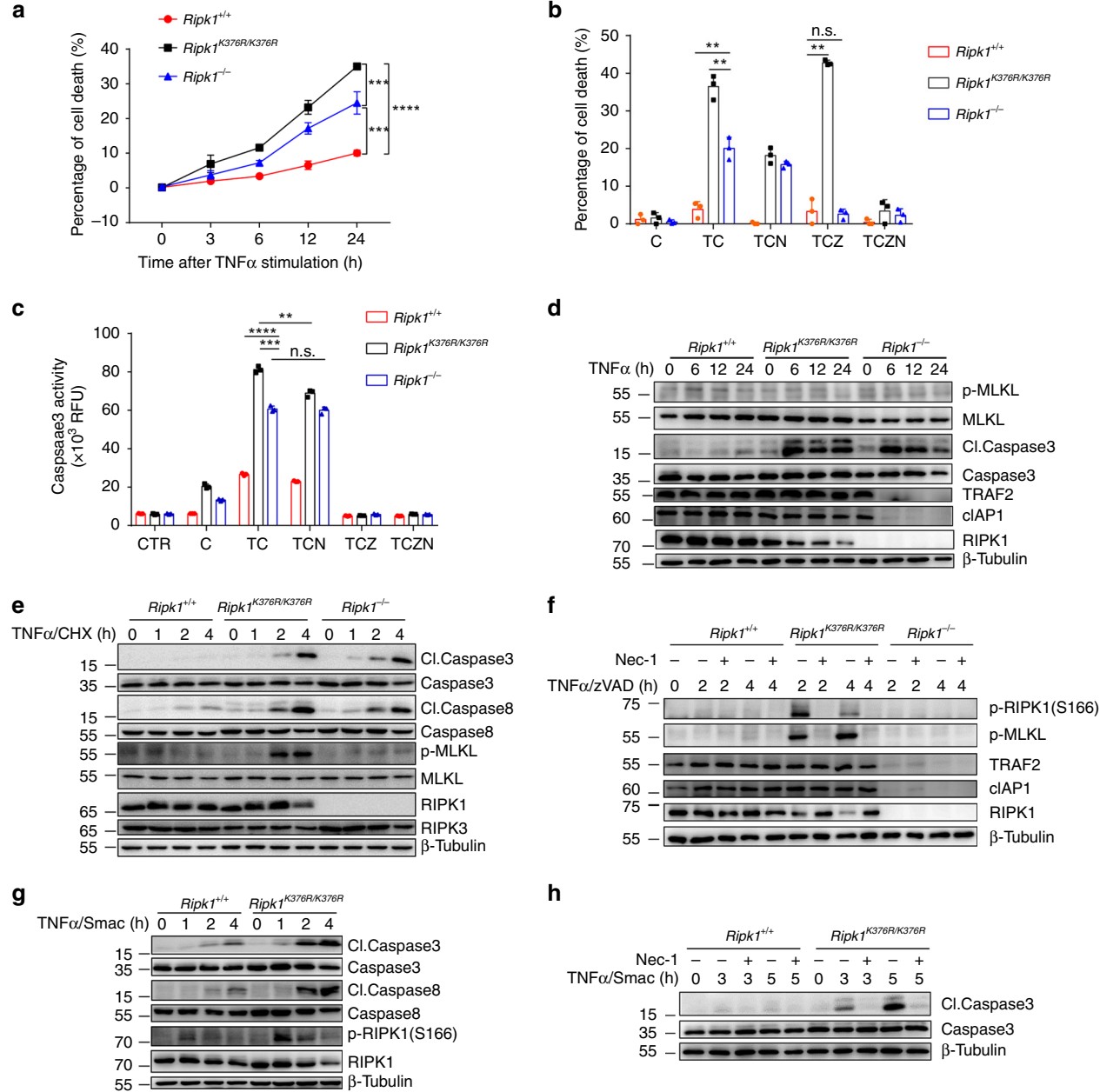

**Fig. 2** *Ripk1^{K376R/K376R}* mutation sensitizes cells to apoptosis and necroptosis. **a** Cell death of immortalized *Ripk1^{+/+}*, *Ripk1^{K376R/K376R}*, and *Ripk1^{−/−}* MEFs treated with TNFα for different time point was measured by SytoxGreen positivity. **b** Cell death of immortalized *Ripk1^{+/+}*, *Ripk1^{K376R/K376R}*, and *Ripk1^{−/−}* MEFs treated for 5 h with different stimulators was measured by SytoxGreen positivity. T, TNFα; C: cycloheximide (CHX); Z: zVAD.fmk; N: necrostatin-1. **c** Caspase3 activity of *Ripk1^{+/+}*, *Ripk1^{K376R/K376R}*, and *Ripk1^{−/−}* immortalized MEFs treated for 5 h with different stimulators were measured by DEVD-AMC fluorescence. **d–h** *Ripk1^{+/+}*, *Ripk1^{K376R/K376R}*, and *Ripk1^{−/−}* immortalized MEFs were treated with TNFα (**d**), TNFα/CHX (**e**), TNFα/ zVAD.fmk (**f**), or TNFα/Smac mimetics (**g**, **h**) with or without pre-treatment of necrostatin-1 for the indicated time, and the cell lysates were analyzed by western blotting using the indicated antibodies. In **a–h**, TNFα, 20 ng/ml; CHX: 10 μg/ml; zVAD.fmk, 20 μM; Smac mimetics, 5 μM; necrostatin-1, 10 μM. In **a–c**, data are mean ± s.e.m. (*n* = 3 independent cell samples for each genotype). Statistical significance was determined using a two-tailed unpaired *t* test, n.s., *P* > 0.05, **P* < 0.01, ***P* < 0.001, and *****P* < 0.0001

accelerate TNFα-induced apoptosis. TNFα/CHX treatment not only induced more cleaved Caspase3/8 in *Ripk1^{K376R/K376R}* MEFs but also induced higher level of phosphorylated mixed lineage kinase domain-like pseudokinase (MLKL), a biomarker for necroptosis (Fig. 2e). Previous studies showed that phosphorylation of RIPK1 on S166 could induce RIPK1 kinase activity, and further promotes the phosphorylation of downstream RIPK3 and MLKL to activate RIPK1-dependent necroptosis[34,35]. With TNFα/zVAD stimulation, phosphorylation of RIPK1 on S166 and phosphorylation of MLKL were all significantly increased in *Ripk1^{K376R/K376R}* MEFs (Fig. 2f).

These phosphorylation events can be fully prevented when treated with Nec-1 (Fig. 2f). Smac is a mitochondrial protein that can be released into the cytosol to promote caspase activation in the cytochrome *c*/Apaf-1/Caspase9 pathway[36]. Consistent with above results, *Ripk1^{K376R/K376R}* MEFs exhibited more cleaved Caspase3/8 and more phosphorylation of RIPK1 under the condition with Smac mimetics plus TNFα stimulation (Fig. 2g). Interestingly, Nec-1 can significantly inhibit Caspase3 activity and cell death in *Ripk1^{K376R/K376R}* MEFs (Fig. 2b, c, h and Supplementary Fig. 2d), indicating that RIPK1 kinase activity may contribute to the

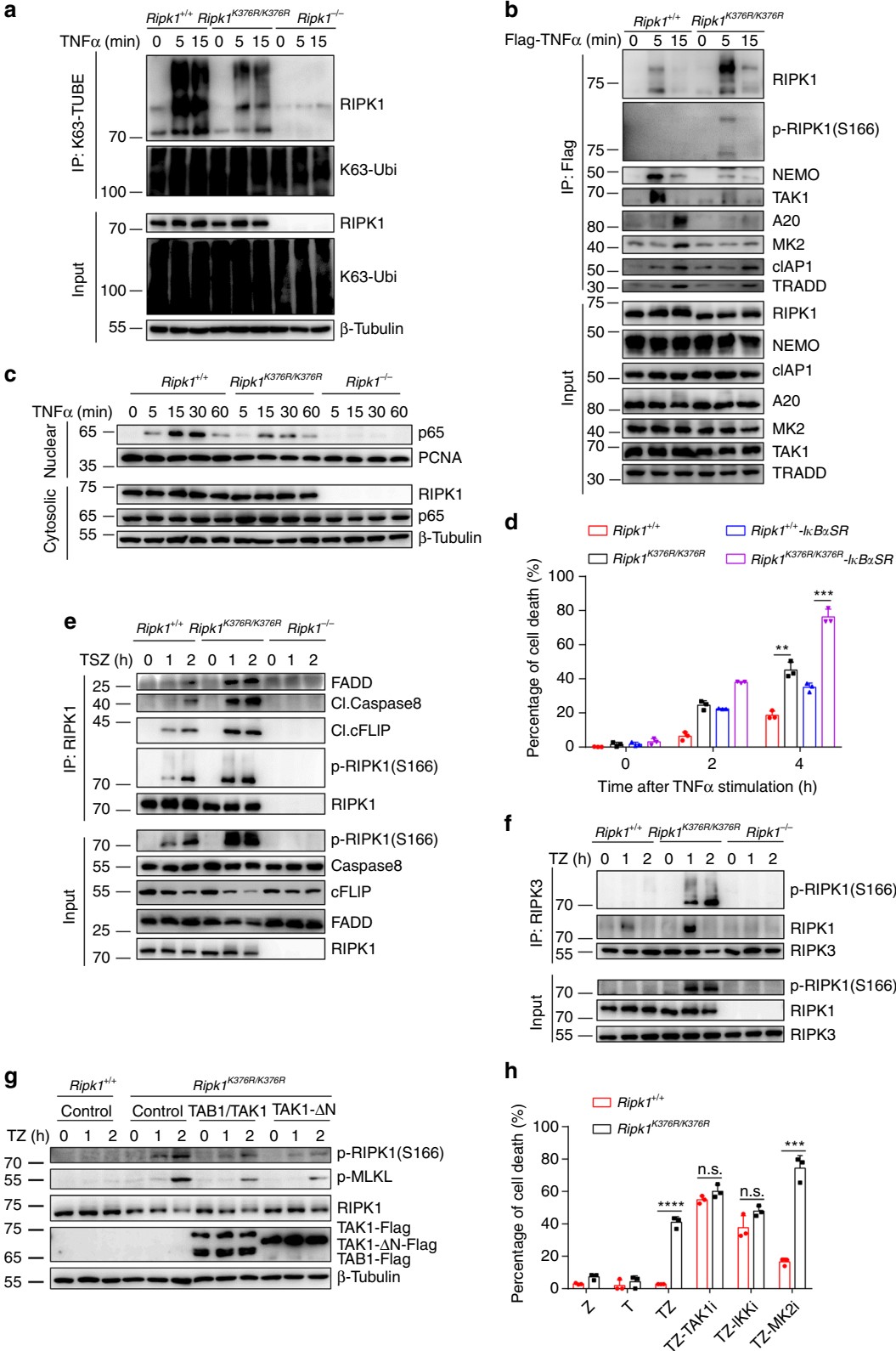

induction of both apoptosis and necroptosis. Together, these data clearly demonstrate that *Ripk1^K376R/K376R* mutation promotes RIPK1-dependent apoptosis and necroptosis.

**_Ripk1^K376R/K376R_ mutation enhances RIPK1 kinase activity.** Previous studies have reported that Lys(K)377 residue is the major K63 ubiquitination site in human RIPK1. To confirm

whether Lys(K)376 residue is the K63 ubiquitination site in murine RIPK1, we detected K63 ubiquitination level of RIPK1 in MEFs. With TNFα stimulation, WT RIPK1 showed increased level of K63-linked ubiquitination, but *Ripk1^K376R/K376R* mutation significantly reduced the K63-linked ubiquitination of RIPK1 (Fig. 3a and Supplementary Fig. 3a), and it has no effect on Met1-linked ubiquitination chains of RIPK1 (Supplementary Fig. 3b).

**Fig. 3** $Ripk1^{K376R/K376R}$ mutation increases RIPK1 kinase activity. **a** Immortalized $Ripk1^{+/+}$, $Ripk1^{K376R/K376R}$, and $Ripk1^{-/-}$ MEFs were treated for the indicated time with TNFα (20 ng/ml). The K63-ubiquitylated proteins were isolated by K63-TUBEs and analyzed by western blot. **b** TNF-RSC was immunoprecipitated using anti-Flag resin in $Ripk1^{+/+}$ and $Ripk1^{K376R/K376R}$ immortalized MEFs treated for the indicated time with Flag-TNFα (100 ng/ml). The immunocomplexes were analyzed by western blot with indicated antibodies. **c** Nuclear extracts were collected from $Ripk1^{+/+}$, $Ripk1^{K376R/K376R}$, and $Ripk1^{-/-}$ immortalized MEFs treated with TNFα (20 ng/ml) at indicated times and analyzed by western blotting with antibodies against p65 and PCNA. **d** $Ripk1^{+/+}$ and $Ripk1^{K376R/K376R}$ immortalized MEFs that stably expressed with the Flag-tagged IκBα-SR were stimulated with TNFα (40 ng/ml) for different periods of time. Cell death was measured by SytoxGreen positivity. **e, f** Complex II was immunoprecipitated using RIPK1 antibody (**e**) or RIPK3 antibody (**f**) in $Ripk1^{+/+}$, $Ripk1^{K376R/K376R}$, and $Ripk1^{-/-}$ immortalized MEFs treated for the indicated time with TSZ (**e**) or TZ (**f**). The immunocomplexes were analyzed by western blotting with indicated antibodies. T, TNFα (20 ng/ml); S, Smac mimetics (1 μM); Z, zVAD.fmk (1 μM). **g** $Ripk1^{+/+}$ and $Ripk1^{K376R/K376R}$ immortalized MEFs that expressed with the Flag-tagged TAK1/TAB1 or TAK1-ΔN were stimulated with TNFα (20 ng/ml) plus zVAD.fmk (10 μM) for different periods of time. The cell lysates were analyzed by western blotting with indicated antibodies. **h** Cell death of $Ripk1^{+/+}$ and $Ripk1^{K376R/K376R}$ immortalized MEFs treated for 5 h with different stimulators was measured by SytoxGreen positivity. T: TNFα (20 ng/ml), Z: zVAD.fmk (10 μM), TAK1i: TAK1 inhibitor (500 μM), IKKi: IKK inhibitor(5 μM), MK2i: MK2 inhibitor (2 μM). In **d, h**, data are mean ± s.e.m. ($n = 3$ independent cell samples for each genotype). Statistical significance was determined using a two-tailed unpaired $t$ test, n.s., $P > 0.05$, **$P < 0.01$, ***$P < 0.001$, and ****$P < 0.0001$

Since K63-linked ubiquitination of RIPK1 in Complex I can function as docking sites for the recruitment of NEMO and TAK1 to further induce NF-κB activation, we examined the formation of Complex I, and found that $Ripk1^{K376R/K376R}$ mutation significantly decreased the recruitment of NEMO, A20, TAK1, and MK2 in Complex I, while it did not affect the recruitment of cIAP1 and TRADD (tumor necrosis factor receptor type 1-associated DEATH domain) (Fig. 3b). Since $Ripk1^{K376R/K376R}$ mutation disrupted the formation of Complex I, we next checked NF-κB activation, and found that p65 translocation was decreased in $Ripk1^{K376R/K376R}$ MEFs (Fig. 3c). We also detected the expression of several NF-κB-targeting genes, and they all showed lower expression level in $Ripk1^{K376R/K376R}$ MEFs (Supplementary Fig. 3c).

Because NF-κB activation could inhibit cell death by inducing anti-apoptotic gene expression, it may explain that the severe cell death of $Ripk1^{K376R/K376R}$ cells was resulted from decreased NF-κB activation. Therefore, we expressed an IκBα super-repressor in MEFs, which could completely block the degradation of IκBα and inhibit NF-κB activation (Supplementary Fig. 3d). However, we found that $Ripk1^{K376R/K376R}$ MEFs were still more sensitive to TNFα-induced cell death (Fig. 3d and Supplementary Fig. 3e), indicating that the increased cell death in $Ripk1^{K376R/K376R}$ MEFs is not only due to its defective NF-κB activation. RIPK1 kinase activity was elevated in $Ripk1^{K376R/K376R}$ MEFs (Fig. 3b), and RIPK1 kinase activity can mediate the transition from Complex I to Complex II. Therefore, we hypothesized that K63-linked ubiquitination of RIPK1 may directly prevent Complex II formation. Consistent with our hypothesis, $Ripk1^{K376R/K376R}$ mutation enhanced the interaction between RIPK1 and FADD, cleaved Caspase8 or cFLIP (cellular FLICE (FADD-like IL-1β-converting enzyme)-inhibitory protein) (Fig. 3e). We could also observe elevated level of phosphorylated RIPK1 on S166 in $Ripk1^{K376R/K376R}$ MEFs (Fig. 3e). Furthermore, $Ripk1^{K376R/K376R}$ mutation increased phosphorylated RIPK1 and promoted the interaction between RIPK1 and RIPK3 in necrosome (Fig. 3f).

Consistent with previous studies, RIPK1 deficiency led to TNFα-induced loss of TRAF2 and cIAP1[37,38]; however, K376 mutation of RIPK1 did not affect TRAF2 and cIAP1 stability (Fig. 2d, f). Since TAK1 could suppress RIPK1 kinase activity by directly phosphorylating RIPK1 or recruiting IKKα/β and MK2 to phosphorylate RIPK1[10–13,39], we hypothesized that TAK1 may contribute to RIPK1 activation in $Ripk1^{K376R/K376R}$ cells. We found only overexpression of TAK1 could not inhibit RIPK1 kinase activity in $Ripk1^{K376R/K376R}$ MEFs (Supplementary Fig. 3f), but overexpression of TAK1/TAB1 complex or a constitutively active mutant of TAK1 (TAK1-ΔN)[40] could significantly reduce the RIPK1 kinase activity (Fig. 3g). Moreover, only treatment with TAK1 inhibitor or IKK inhibitor diminished the difference

of cell death and RIPK1 kinase activity between WT and $Ripk1^{K376R/K376R}$ MEFs, although MK2 inhibitor can increase cell death in both WT and $Ripk1^{K376R/K376R}$ MEFs (Fig. 3h and Supplementary Fig. 3g–i). Similarly, overexpression of constitutively active IKKβ (IKKβ-CA) but not MK2 suppress RIPK1 kinase activity in $Ripk1^{K376R/K376R}$ MEFs (Supplementary Fig. 3j, k). Furthermore, the interaction between TAK1 and RIPK1 was decreased in $Ripk1^{K376R/K376R}$ MEFs (Supplementary Fig. 3l), indicating that loss of RIPK1 K376 ubiquitination prevents TAK1 recruitment. K376 mutation of RIPK1 also inhibited TAK1 downstream MAPK activation (Supplementary Fig. 3m). These data suggested that K376R mutation in RIPK1 suppress RIPK1 kinase activity mainly by inhibiting TAK1-IKK axis activity. Taken together, K63-linked ubiquitination of RIPK1 on K376 not only promotes NF-κB activation by activating Complex I but also regulates TAK1-IKK activity to suppress RIPK1 kinase activity that prevents Complex II formation, thereby inhibiting cell death.

**TNFR1-dependent signaling partially causes the lethality.** It has been shown that ablation of TNFR1 delays, but cannot rescue, the lethality of $Ripk1^{-/-}$ mice[32,33,41]. Since $Ripk1^{K376R/K376R}$ mutation promotes TNFα-induced cell death, we determine the impact of TNFR1 deficiency on the lethality of $Ripk1^{K376R/K376R}$ mice. Similar to $Ripk1^{-/-}Tnfr1^{-/-}$ mice, $Ripk1^{K376R/K376R}Tnfr1^{-/-}$ mice were born at Mendelian frequencies (Supplementary Fig. 4a), but also died around postnatal day 12 (Fig. 4a). Remarkably, $Ripk1^{K376R/K376R}Tnfr1^{-/-}$ mice developed severe systemic inflammation around 10 days old, and showed reduced body size and smaller organs, including thymus and heart (Fig. 4b and Supplementary Fig. 4b). Histological staining showed that $Ripk1^{K376R/K376R}Tnfr1^{-/-}$ mice had more infiltrated inflammatory cells in the liver region, and the skin epidermal thickness was significantly increased (Fig. 4c). We could also observe more cleaved Caspase3 and infiltrated immune cells in the skin sections of $Ripk1^{K376R/K376R}Tnfr1^{-/-}$ mice than those in $Ripk1^{K376R/+}Tnfr1^{-/-}$ mice (Fig. 4d and Supplementary Fig. 4c). Moreover, the expression level of skin differentiation markers was significantly increased in the $Ripk1^{K376R/K376R}Tnfr1^{-/-}$ mice, suggesting that $Ripk1^{K376R/K376R}$ mutation promotes cell death to further regulate skin inflammation (Fig. 4e). Furthermore, more cleaved Caspase3 and CD11b could be found in the liver region of $Ripk1^{K376R/K376R}Tnfr1^{-/-}$ mice (Fig. 4f). Interestingly, $Ripk1^{-/-}Tnfr1^{-/-}$ mice displayed almost the same phenotype as $Ripk1^{K376R/K376R}Tnfr1^{-/-}$ mice (Supplementary Fig. 4d–g), indicating that K376R mutation of RIPK1 is similar to the RIPK1 deficiency that induces inflammation when TNFR1 signaling was blocked.

Liver injury is one of the key reason for the embryonic lethality in IKKβ- or NEMO-deficient mice, which have NF-κB activation

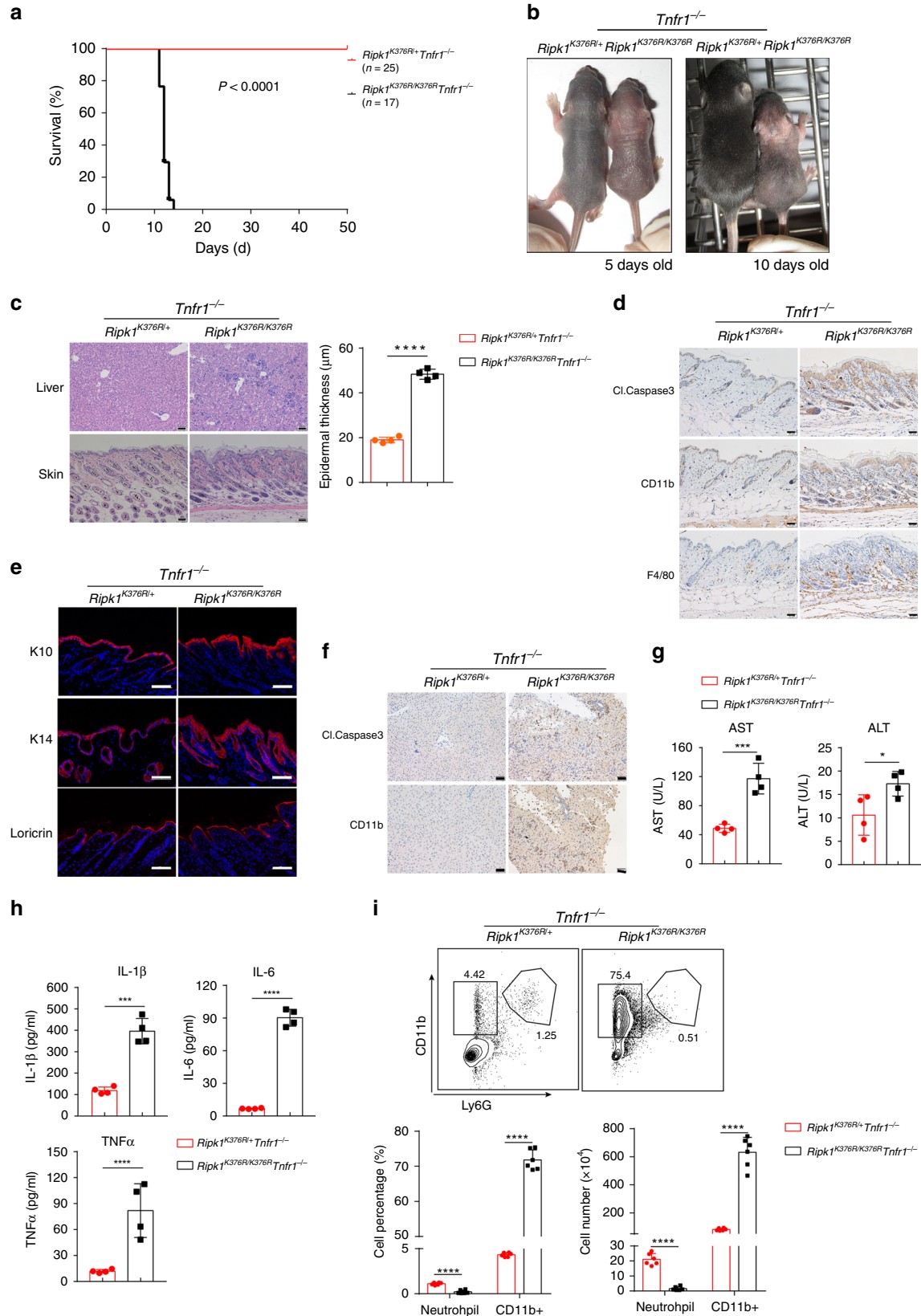

defect[42,43]. Thus, we examined the level of aspartate transaminase/alanine transaminase (AST/ALT), which is the hallmark of liver injury. We found that AST/ALT were significantly increased in the serum of $Ripk1^{K376R/K376R}Tnfr1^{-/-}$ mice (Fig. 4g). The inflammatory cytokines, including IL-1β, IL-6, and TNFα, were all upregulated in the liver and lung of $Ripk1^{K376R/K376R}Tnfr1^{-/-}$ mice (Fig. 4h and Supplementary Fig. 4h). Finally, we analyzed the composition of immune cells in the spleen. The total splenocytes, CD4$^+$ T cells, CD8$^+$ T cells, and B cells were significantly decreased in the spleen of $Ripk1^{K376R/K376R}Tnfr1^{-/-}$

**Fig. 4** TNFR1 deficiency partially delays the lethality of $Ripk1^{K376R/K376R}$ mice. **a** Survival curves of pups from intercrosses of $Ripk1^{K376R/+}Tnfr1^{-/-}$ mice. **b** Representative macroscopic images of $Ripk1^{K376R/K376R}Tnfr1^{-/-}$ and $Ripk1^{K376R/+}Tnfr1^{-/-}$ littermate mice at P5 and P10. **c** H&E staining of liver and skin sections of $Ripk1^{K376R/K376R}Tnfr1^{-/-}$ and $Ripk1^{K376R/+}Tnfr1^{-/-}$ littermate mice at P11 (scale bar, 50 μm), and quantification of the epidermal thickness from H&E results ($Ripk1^{K376R/+}Tnfr1^{-/-}$ mice: $n = 4$; $Ripk1^{K376R/K376R}Tnfr1^{-/-}$ mice: $n = 4$). **d, e** Immunohistochemical staining of F4/80, CD11b, and cleaved Caspase3 (scale bar, 50 μm) (**d**) or immunofluorescence staining of Loricrin, K10, and K14 (scale bar, 100 μm) (**e**) in skin sections of $Ripk1^{K376R/K376R}$ $Tnfr1^{-/-}$ and $Ripk1^{K376R/+}Tnfr1^{-/-}$ littermate mice at P11. **f** Immunohistochemical staining of CD11b and cleaved Caspase3 staining in liver sections of $Ripk1^{K376R/K376R}Tnfr1^{-/-}$ and $Ripk1^{K376R/+}Tnfr1^{-/-}$ littermate mice at P11 (scale bar, 50 μm). **g** AST and ALT in blood were determined with the indicated genotypes at P11 ($Ripk1^{K376R/+}Tnfr1^{-/-}$ mice: $n = 4$; $Ripk1^{K376R/K376R}Tnfr1^{-/-}$ mice: $n = 4$). **h** Cytokines in liver homogenates were determined with the indicated genotypes at P11 ($n = 4$ biologically independent mice). **i** Flow cytometry and statistical results of splenocytes stained with Ly6G and CD11b from $Ripk1^{K376R/K376R}Tnfr1^{-/-}$ and $Ripk1^{K376R/+}Tnfr1^{-/-}$ littermate mice at P11 ($Ripk1^{K376R/+}Tnfr1^{-/-}$ mice: $n = 6$; $Ripk1^{K376R/K376R}Tnfr1^{-/-}$ mice: $n = 6$). CD11b$^+$Ly6G$^+$ cells were identified as neutrophils. In **c, g–i**, data are mean ± s.e.m. Statistical significance was determined using a two-tailed unpaired $t$ test, *$P < 0.05$, ***$P < 0.001$, and ****$P < 0.0001$

mice, suggesting that $Ripk1^{K376R/K376R}$ mutation can promote cell death in a TNFR1-independent pathway to further affect the development of immune cells (Supplementary Fig. 4i). However, the inflammatory CD11b$^+$ cells were largely increased (Fig. 4i), which was consistent with the inflammatory phenotype observed in $Ripk1^{K376R/K376R}Tnfr1^{-/-}$ mice. Besides TNFα, RIPK1 is also involved in the signaling induced by TLRs and interferons[44,45]. We found that $Ripk1^{K376R/K376R}$ MEFs were more sensitive to lipopolysaccharide- or IFNγ-induced cell death (Supplementary Fig. 5a, b). TNF-related apoptosis-inducing ligand (TRAIL)- and Fas ligand (FasL)-induced cell death was also increased in $Ripk1^{K376R/K376R}$ MEFs (Supplementary Fig. 5c–e). Together, these data suggest that TNFR1-mediated signaling is partially responsible for the embryonic lethality and inflammation of $Ripk1^{K376R/K376R}$ mice.

**Cell death is responsible for the lethality and inflammation.** To address whether the cell death signaling led to the lethality of $Ripk1^{K376R/K376R}$ mice, we crossed RIPK3- and Caspase8-double-KO mice with $Ripk1^{K376R/K376R}$ mice. The $Ripk1^{K376R/K376R}$ $Ripk3^{-/-}Caspase8^{-/-}$ mice were viable and fertile and had no obvious morphological difference compared to their $Ripk1^{K376R/+}Ripk3^{-/-}Caspase8^{-/-}$ littermate controls (Fig. 5a), indicating that the double deficiency of RIPK3 and Caspase8 rescues the cell death in $Ripk1^{K376R/K376R}$ mice. The H&E staining results showed no inflammation in the liver region and the skin thickness were normal (Fig. 5b). Cleaved Caspase3 was absent and the infiltrated immune cells had no difference between $Ripk1^{K376R/K376R}Ripk3^{-/-}Caspase8^{-/-}$ mice and $Ripk1^{K376R/+}$ $Ripk3^{-/-}Caspase8^{-/-}$ littermate controls (Fig. 5c and Supplementary Fig. 6a). In addition, the rescued mice had normal skin development as shown by staining of K14, K10, and loricrin (Fig. 5d). Furthermore, the expression level of AST and ALT in serum and inflammatory cytokines in liver were all in the normal level in the rescued mice (Fig. 5e, f). We also analyzed the composition of immune cells in the spleen, and the results showed that CD4$^+$ T cells, CD8$^+$ T cells, B cells, CD11b$^+$ cells, and neutrophils (CD11b$^+$Ly6G$^+$) displayed no significant difference between rescued mice and littermate controls (Fig. 5g and Supplementary Fig. 6b). However, only RIPK3 deficiency cannot rescue the lethality in $Ripk1^{K376R/K376R}$ mice (Supplementary Fig. 6c). Together, these data indicate that blocking RIPK3-dependent necrosis and Caspase8-dependent apoptosis can fully prevent the lethality and systemic inflammation in $Ripk1^{K376R/K376R}$ mice.

**$Ripk1^{K376R/-}$ mice develop severe systemic inflammation.** $Ripk1^{K376R/K376R}$ mice were embryonically lethal, which limited us to further investigate the physiological function of K63-linked ubiquitination of RIPK1 in vivo. Surprisingly, when crossing $Ripk1^{+/-}$ mice with $Ripk1^{K376R/+}$ mice, we got viable $Ripk1^{K376R/-}$

mice. $Ripk1^{K376R/-}$ mice were weaned at the Mendelian ratio (Supplementary Fig. 7a), but developed severe systemic inflammation after birth, characterized by dermatitis, extensive spleno-megaly, and reduced body size (Fig. 6a and Supplementary Fig. 7b–d). The severe inflammation in $Ripk1^{K376R/-}$ mice could result in the lethality at different stages (Fig. 6b). In addition, histological staining showed that $Ripk1^{K376R/-}$ mice had massive infiltration of lymphocytes in the liver and significantly thickened skin epidermis (Fig. 6c). Immunohistological staining of skin sections showed that higher level of cleaved Caspase3, indicating more cell apoptosis, and also the infiltrated immune cells were largely increased in $Ripk1^{K376R/-}$ mice (Fig. 6d and Supplementary Fig. 7e). $Ripk1^{K376R/-}$ mice had abnormal skin lesion, indicated by elevated expression level of K10, K14, and loricrin (Fig. 6e). Furthermore, we detected higher levels of AST in the serum, and increased inflammatory cytokines in the liver, indicating the severe liver injury and inflammation in $Ripk1^{K376R/-}$ mice (Supplementary Fig. 7f). Although the total number of splenocytes was comparable, we observed significantly decreased CD4$^+$ T cells, CD8$^+$ T cells, B cells, and largely increased neutrophils in the spleen of $Ripk1^{K376R/-}$ mice, which confirmed the severe systemic inflammation of $Ripk1^{K376R/-}$ mice (Fig. 6f and Supplementary Fig. 7g). $Ripk1^{K376R/K376R}$ mutation could promote cell death and further result in embryonic lethality, and $Ripk1^{K376R/-}$ mice only have half K376R mutation allele compared with $Ripk1^{K376R/K376R}$ mice. Therefore, we proposed that $Ripk1^{K376R/-}$ mutation reduced the sensitivity to cell death compared to $Ripk1^{K376R/K376R}$ mutation, because of the decreased dose of RIPK1 kinase activity. Consistent with our hypothesis, $Ripk1^{K376R/-}$ MEFs were more sensitive to TNFα-induced apoptosis and necroptosis than WT control, but had lower sensitivity to cell death compared to $Ripk1^{K376R/K376R}$ MEFs (Fig. 6g and Supplementary Fig. 7h). Western blotting analysis showed the similar results that $Ripk1^{K376R/-}$ MEFs had lower level of cleaved Caspase3 and phosphorylation of MLKL, especially the reduced RIPK1 kinase activity compared to $Ripk1^{K376R/K376R}$ MEFs (Fig. 6h, i and Supplementary Fig. 7i, j). To rule out the effect of truncated RIPK1 expression in $Ripk1^{K376R/-}$ mice, we also generated another strain of $Ripk1^{-/-}$ mice (indicated by $KD$-$Ripk1^{-/-}$ mice), in which the translation of RIPK1 stopped at Valine 42 (V42) in the kinase domain and cannot detect RIPK1 expression in $KD$-$Ripk1^{-/-}$ MEFs (Supplementary Fig. 8a, b). $KD$-$Ripk1^{K376R/-}$ mice displayed the similar phenotypes as $Ripk1^{K376R/-}$ mice (Supplementary Fig. 8c–f). Together, these results indicated that $Ripk1^{K376R/-}$ mice developed severe systemic inflammation driven by relatively decreased cell death compared to $Ripk1^{K376R/K376R}$ mice.

**TNFα mainly drives the inflammation of $Ripk1^{K376R/-}$ mice.** To examine the contribution of TNFR1-mediated signaling to the

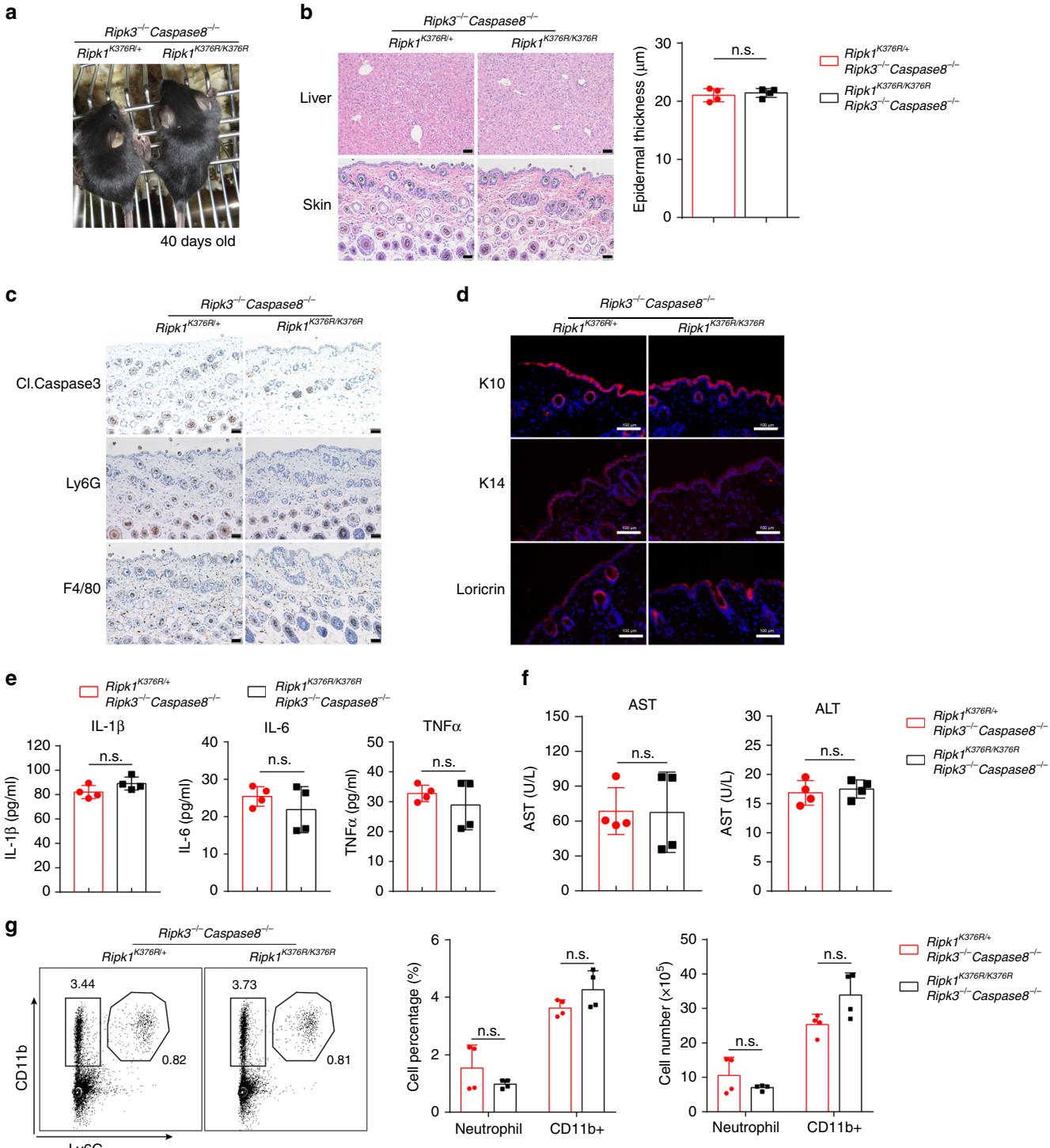

**Fig. 5** Co-deletion of RIPK3 and Caspase8 fully rescue *Ripk1*$^{K376R/K376R}$ mice. **a** Representative macroscopic images of *Ripk1*$^{K376R/K376R}$*Ripk3*$^{−/−}$*Caspase8*$^{−/−}$ and *Ripk1*$^{K376R/+}$*Ripk3*$^{−/−}$*Caspase8*$^{−/−}$ littermate mice at P40. **b** H&E staining of liver and skin sections of *Ripk1*$^{K376R/K376R}$*Ripk3*$^{−/−}$ *Caspase8*$^{−/−}$ and *Ripk1*$^{K376R/+}$*Ripk3*$^{−/−}$*Caspase8*$^{−/−}$ littermate mice at P40 (scale bar, 50 μm), and microscopic quantification of the epidermal thickness from H&E results (*Ripk1*$^{K376R/+}$*Ripk3*$^{−/−}$*Caspase8*$^{−/−}$ mice: *n* = 4; *Ripk1*$^{K376R/K376R}$*Ripk3*$^{−/−}$ *Caspase8*$^{−/−}$ mice: *n* = 4). **c, d** Immunohistochemical staining of F4/80, Ly6G, and cleaved Caspase3 (scale bar, 50 μm) (**c**) or immunofluorescence staining of Loricrin, K10, and K14 (scale bar, 100 μm) (**d**) in skin sections of *Ripk1*$^{K376R/K376R}$*Ripk3*$^{−/−}$*Caspase8*$^{−/−}$ and *Ripk1*$^{K376R/+}$*Ripk3*$^{−/−}$*Caspase8*$^{−/−}$ littermate mice at P40. **e** Cytokines in liver homogenates were determined with the indicated genotypes at P40 (*Ripk1*$^{K376R/+}$*Ripk3*$^{−/−}$*Caspase8*$^{−/−}$ mice: *n* = 4; *Ripk1*$^{K376R/K376R}$*Ripk3*$^{−/−}$ *Caspase8*$^{−/−}$ mice: *n* = 4). **f** AST and ALT in blood were determined with the indicated genotypes at P40 (*Ripk1*$^{K376R/+}$*Ripk3*$^{−/−}$*Caspase8*$^{−/−}$ mice: *n* = 4; *Ripk1*$^{K376R/K376R}$*Ripk3*$^{−/−}$ *Caspase8*$^{−/−}$ mice: *n* = 4). **g** Flow cytometry and statistical results of splenocytes stained with Ly6G and CD11b from *Ripk1*$^{K376R/K376R}$*Ripk3*$^{−/−}$*Caspase8*$^{−/−}$ and *Ripk1*$^{K376R/+}$ *Ripk3*$^{−/−}$*Caspase8*$^{−/−}$ littermate mice at P40 (*Ripk1*$^{K376R/+}$*Ripk3*$^{−/−}$*Caspase8*$^{−/−}$ mice: *n* = 4; *Ripk1*$^{K376R/K376R}$*Ripk3*$^{−/−}$ *Caspase8*$^{−/−}$ mice: *n* = 4). CD11b$^+$Ly6G$^+$ cells were identified as neutrophils. In **b**, **e–g**, data are mean ± s.e.m. Statistical significance was determined using a two-tailed unpaired *t* test, n.s., *P* > 0.05

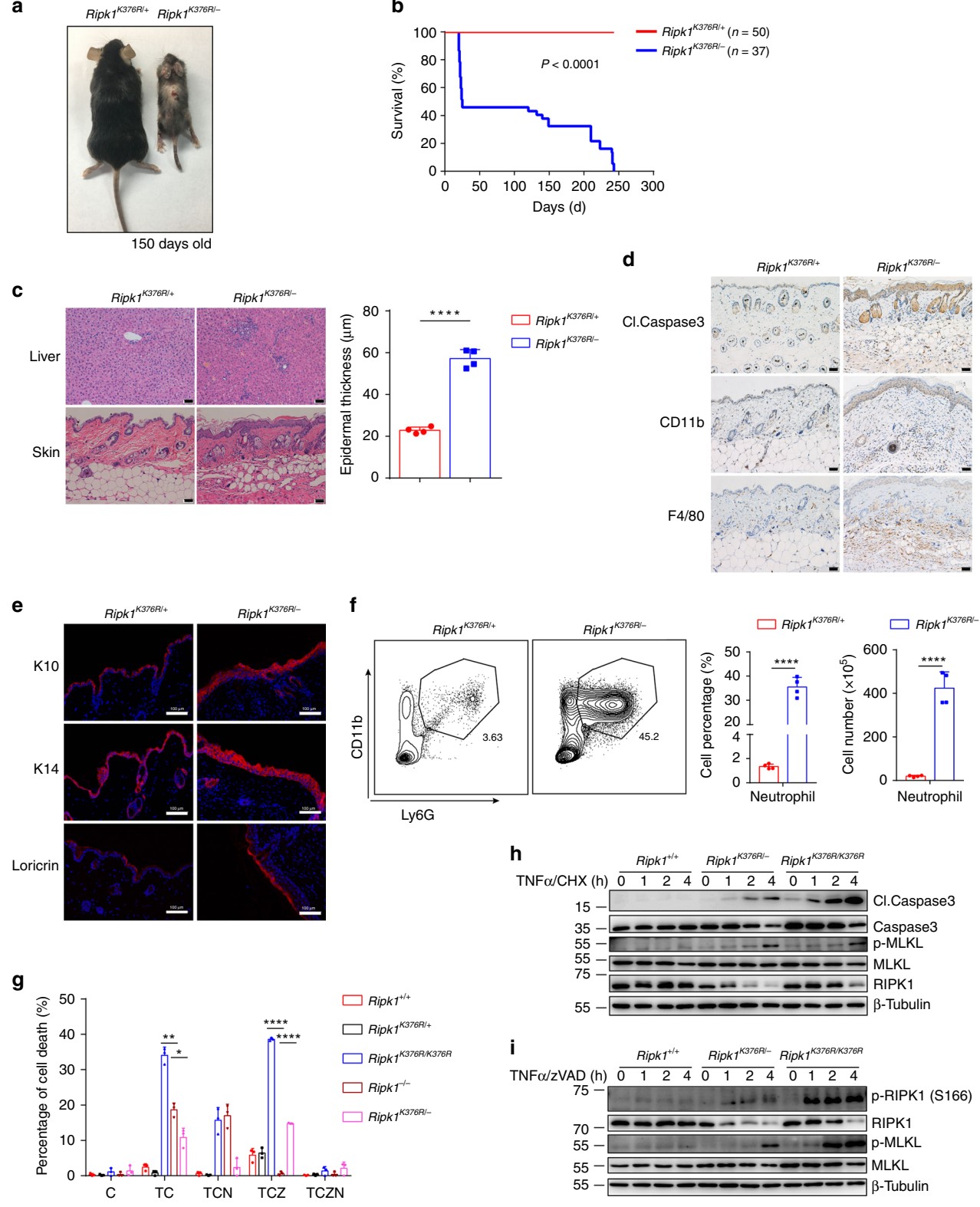

inflammation of $Ripk1^{K376R/-}$ mice, we generated $Ripk1^{K376R/-}$ $Tnfr1^{-/-}$ mice. The $Ripk1^{K376R/-}Tnfr1^{-/-}$ mice survived normally and had no obvious inflammation features (Fig. 7a). The decreased body weight, thickened skin epidermis, more infiltrated massive lymphocytes in the liver or skin, severe liver injury, upregulated level of cleaved Caspase3, and skin differentiation markers were all rescued to normal condition in $Ripk1^{K376R/-}$

$Tnfr1^{-/-}$ mice (Fig. 7b, c and Supplementary Fig. 9a–d). Although most of the inflammatory features disappeared, there was a slight increase in neutrophil recruitment and decreased CD4[+] T cells in the spleen of $Ripk1^{K376R/-}Tnfr1^{-/-}$ mice (Fig. 7d and Supplementary Fig. 9e). This suggested that TNFR1 deficiency could not fully rescue the inflammation of $Ripk1^{K376R/-}$ mice. Next, we generated $Ripk1^{K376R/-}Ripk3^{-/-}Caspase8^{-/-}$

**Fig. 6** $Ripk1^{K376R/-}$ mice develops spontaneous inflammation. **a** Representative macroscopic images of $Ripk1^{K376R/-}$ and $Ripk1^{K376R/+}$ littermate mice at P150. **b** Survival curves of $Ripk1^{K376R/-}$ and $Ripk1^{K376R/+}$ mice. **c** H&E staining of liver and skin sections of $Ripk1^{K376R/-}$ and $Ripk1^{K376R/+}$ littermate mice at P150 (scale bar, 50 μm), and microscopic quantification of the epidermal thickness from H&E results ($Ripk1^{K376R/+}$ mice: $n = 4$; $Ripk1^{K376R/-}$ mice: $n = 4$). **d**, **e** Immunohistochemical staining of cleaved Caspase3, CD11b, and F4/80 (scale bar, 50 μm) (**d**) or immunofluorescence staining of Loricrin, K10, and K14 (scale bar, 100 μm) (**e**) in skin sections of $Ripk1^{K376R/-}$ and $Ripk1^{K376R/+}$ littermate mice at P150. **f** Flow cytometry and statistical results of splenocytes stained with Ly6G and CD11b from $Ripk1^{K376R/-}$ and $Ripk1^{K376R/+}$ littermate mice at P150 ($Ripk1^{K376R/+}$ mice: $n = 4$; $Ripk1^{K376R/-}$ mice: $n = 4$). CD11b$^+$ Ly6G$^+$ cells were identified as neutrophils. **g** Cell death of $Ripk1^{+/+}$, $Ripk1^{K376R/+}$, $Ripk1^{K376R/K376R}$, $Ripk1^{-/-}$, and $Ripk1^{K376R/-}$ immortalized MEFs treated for 5 h with different stimulators was measured by SytoxGreen positivity. The error bars represent mean ± s.e.m. of data from three independent cell samples for each genotype. T: TNFα; C: cycloheximide (CHX); Z: zVAD.fmk; N: necrostatin-1. **h**, **i** $Ripk1^{+/+}$, $Ripk1^{K376R/K376R}$, and $Ripk1^{K376R/-}$ immortalized MEFs were treated with TNFα/CHX (**h**) or TNFα/ zVAD.fmk (**i**) for the indicated time, and the cell lysates were analyzed by western blotting using the indicated antibodies. In **g–i**, TNFα, 20 ng/ml; CHX: 10 μg/ml; zVAD.fmk, 20 μM; necrostatin-1, 10 μM. In **c**, **f**, **g**, data are mean ± s.e.m. Statistical significance was determined using a two-tailed unpaired $t$ test, $*P < 0.05$, $**P < 0.01$ and $****P < 0.0001$

mice. All inflammation features in these mice were restored to a normal level, including neutrophil recruitment and CD4$^+$ T cell number (Fig. 7e–g and Supplementary Fig. 10a–f). Remarkably, we found that $Ripk1^{K376R/-}$$Ripk3^{-/-}$ mice had no obvious inflammation as $Ripk1^{K376R/-}$$Ripk3^{-/-}$$Caspase8^{-/-}$ (Supplementary Fig. 11a–d), indicating that RIPK3-dependent signaling pathway mainly triggers the inflammation in $Ripk1^{K376R/-}$ mice. Taken together, these data supported our conclusion that TNFα-induced cell death dominantly drives the systemic inflammation in $Ripk1^{K376R/-}$ mice.

## Discussion

Cell death and inflammation are two closely related processes, and RIPK1 is a key mediator in both cell death and inflammation process. RIPK1 deficiency caused by rare homozygous mutations could result in severe immunodeficiency, arthritis, and intestinal inflammation in human patients[46]. TAK1-mediated suppression on RIPK1 kinase activity can synergize with genetic risk factors to promote neuroinflammation[47]. Understanding the actual regulation of RIPK1 kinase activity will be beneficial for designing new therapeutic strategy for some clinical disease[48,49]. In this study, we provided the direct evidence that K63-linked ubiquitination of RIPK1 on K376 residue can directly suppress RIPK1 kinase activity to inhibit cell death. Surprisingly, $Ripk1^{K376R/K376R}$ mice died around E13.5, which was much earlier than RIPK1-KO mice. This phenomenon suggested that the function of K63-linked ubiquitination of RIPK1 was more critical than the adaptor function of RIPK1. $Ripk1^{K376R/-}$ mice could survive longer than $Ripk1^{K376R/K376R}$ mice due to relatively decreased RIPK1 kinase activity, further indicating that RIPK1 kinase activity was critical for regulating cell death (Supplementary Fig. 12). TNFR1-deficiency only delayed the lethality of $Ripk1^{K376R/K376R}$ mice, but RIPK3/Caspase8 co-deletion fully prevented the lethality and inflammation of $Ripk1^{K376R/K376R}$ mice. These indicated that K376R mutation of RIPK1 also promoted TNFR1-independent cell death pathways; however, further studies should elucidate TNFR1-independent pathway in the regulation of lethality $Ripk1^{K376R/K376R}$ mice.

K63-linked ubiquitination of RIPK1 provides docking sites for the recruitment of TAK1/TAB complex and NEMO, which further activates NF-κB signaling pathway[50,51]. Mice that lose key components in NF-κB activation pathway could be embryonically lethal. For example, IKKβ- or NEMO-deficient mice are embryonically lethal due to severe liver degeneration[42,43], while HOIP and HOIL-1 deficiency causes embryonic lethality at midgestation due to RIPK1-dependent endothelial cell death[24,25]. In our $Ripk1^{K376R/K376R}$ mice, we can observe massive cell death in liver region, which is similar to IKKβ- or NEMO-deficient mice[42,43]. It will be interesting to determine why RIPK1 K376R mutation appears to selectively promote fetal liver cell death.

According to our results, reduced NF-κB activation was not the exact reason for enhanced cell death in $Ripk1^{K376R/K376R}$ MEFs, which was different from previous findings[19,20]. Similarly, intestinal epithelial cell-specific ablation of NEMO promoted cell death and inflammation by a NF-κB-independent manner, and the inflammation can be prevented by inhibition of RIPK1 kinase activity[52]. Further study needs to assess whether enhanced NF-κB activation can rescue lethality of $Ripk1^{K376R/K376R}$ mice in vivo.

We found that K63-linked ubiquitination of RIPK1 on K376 suppresses RIPK1 kinase activity by regulating TAK1/IKK axis activity. Consistent with our conclusion, recent findings showed that TAK1 could directly phosphorylate or recruit IKKα/β to phosphorylate RIPK1 to suppress RIPK1 activation independently of NF-κB activation[10,39]. MK2 is another downstream kinase of TAK1, which could also suppress RIPK1 kinase activity[11–13]. Although we observed decreased recruitment of MK2 in Complex I and decreased phosphorylation of MK2 in $Ripk1^{K376R/K376R}$ cells, unlike TAK1 and IKK inhibitors, MK2 inhibitor can only weakly enhance TNFα/zVAD-induced cell death in WT cells to the level as that in $Ripk1^{K376R/K376R}$ cells. Together, these results suggest that ubiquitination of RIPK1 on K376 mainly recruits TAK1/IKK to suppress RIPK1 kinase activity.

Previous studies have shown that cIAP1/cIAP2 can conjugate K63-linked ubiquitination chains to RIPK1, and $Ciap1/2^{-/-}$ mice die at E10.5[53]. Similarly, $Ripk1^{K376R/K376R}$ mice died around E13.5, confirming that the K63-linked ubiquitination of RIPK1 on K376 was critical for cell survival. A recent study reported that another E3 ligase Parkin can also mediate K63-linked ubiquitination of RIPK1 on K376 to promote NF-κB activation[54]. Besides, PELI1 could mediate K63-linked ubiquitination of RIPK1 on K115 in a kinase-dependent manner during necroptosis[55]. Thus, RIPK1 can produce K63-linked ubiquitination chains at different sites and by different E3 ligases. In addition, K11- and Met1-linked ubiquitination of RIPK1 have a signaling role in the TNFR1-meidated NF-κB activation[56,57]. Interestingly, K63- and Met1-linked ubiquitination have a crosstalk during the regulation of RIPK1 activation[31]. Therefore, different ubiquitination types and different E3 ligases regulating RIPK1 make it complicated for distinguishing the physiological function of RIPK1 ubiquitination status. By generating $Ripk1^{K376R/K376R}$ mice, we defined a previously unexpected regulatory loop of RIPK1 function that K63-linked ubiquitination of RIPK1 on K376 could regulate RIPK1 kinase activity, thereby contributing to the process of embryogenesis and inflammation. These insights may provide a further rationale for exploring the clinical treatment against inflammation and cancers.

## Methods

**Mice**. $Rik1^{K376R/K376R}$ mutation mice were generated by CRISPR-Cas9 technique. Briefly, one single guide RNA (sgRNA) that target the DNA region surrounding

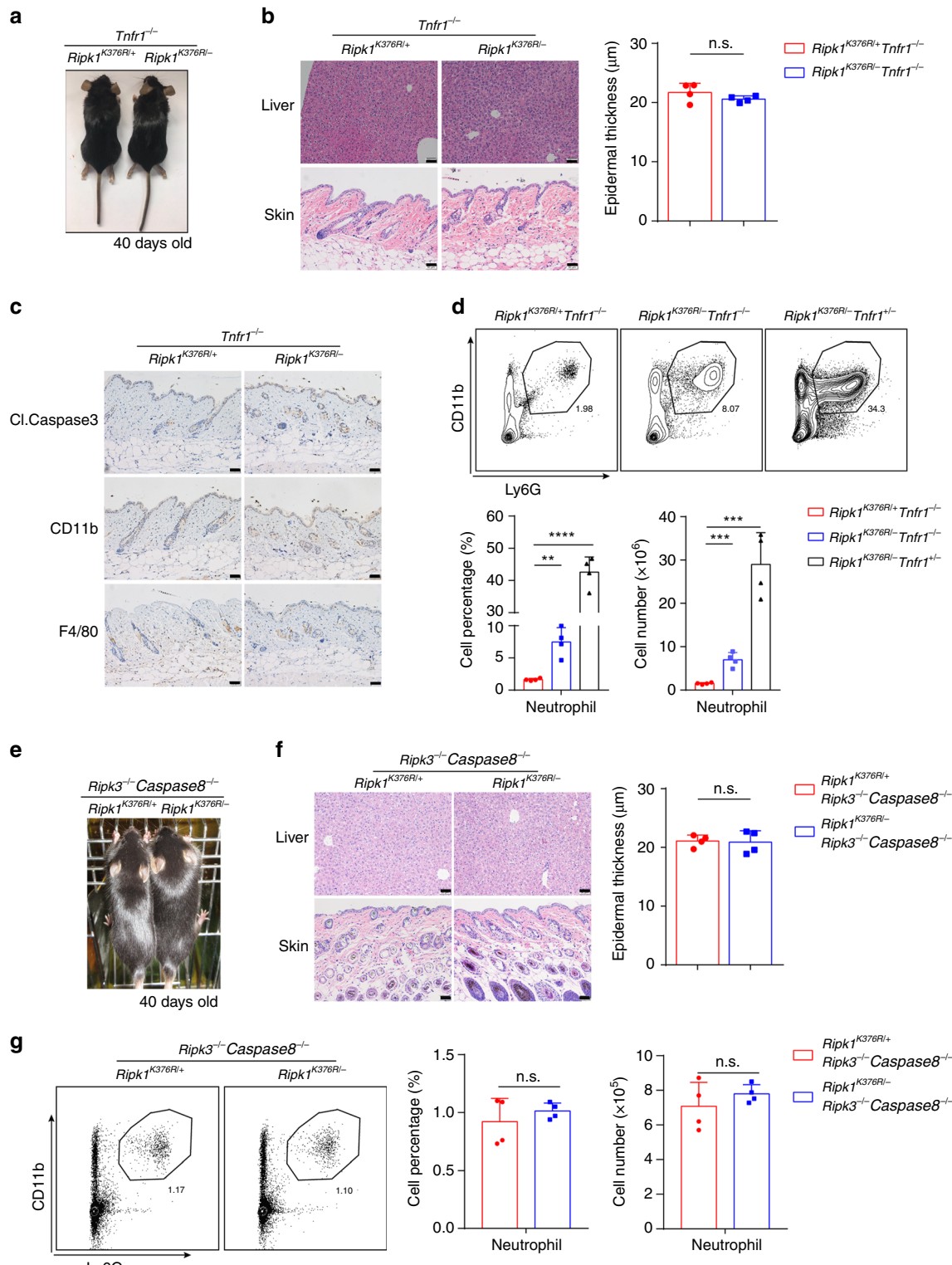

K376 of RIPK1 and the corresponding donor construct (containing K to R mutation) were designed (see in Supplementary Table 1). To prevent the sgRNA-directed Cas9 from cleaving mutated RIPK1 genomic locus, we also constructed several synonymous mutations within the donor. Then, the donor constructs, the mRNA of the sgRNA and Cas9, were injected together into the fertilized eggs of C57BL/6 mice. $KD$-$Ripk1^{-/-}$ mice was also generated by CRISPR-Cas9 technique following the similar process above. The corrected founder mice were genotyped by genomic PCR and DNA sequencing (see in Supplementary Table 2). WT C57BL/6 mice, TNFR1 KO, RIPK3, and Caspase8-KO mice were purchased from Jackson Laboratory and bred in the facility. All mice were housed in the specific pathogen-free animal facilities in Tsinghua University. All mouse experiments were

performed in compliance with institutional guidelines and according to the protocol approved by the Institutional Animal Care and Use Committee of Tsinghua University.

**Antibodies and reagents**. Antibodies against RIPK1 (3493), phospho-RIPK1 (31122), cleaved Caspase3 (9664), Caspase3 (9665), cleaved Caspase8 (9429), Caspase8 (4790), NEMO (2685S), MK2 (3042), Lys63 ubiquitin antibody (5621), A20 (5630), cFLIP (56343), and cIAP1 (4952) were purchased from Cell Signaling Technology; antibodies against RIPK1 (610459) and FasL (554255) was purchased from BD Biosciences; antibodies against RIPK3 (sc-374639), IκBα (sc-1643),

**Fig. 7** TNFR1-mediated cell death triggers the inflammation of $Ripk1^{K376R/−}$ mice. **a** Representative macroscopic images of $Ripk1^{K376R/−}Tnfr1^{−/−}$ and $Ripk1^{K376R/+}Tnfr1^{−/−}$ littermate mice at P40. **b** H&E staining of liver and skin sections of $Ripk1^{K376R/−}Tnfr1^{−/−}$ and $Ripk1^{K376R/+}Tnfr1^{−/−}$ littermate mice at P40 (scale bar, 50 μm), and microscopic quantification of the epidermal thickness from H&E results ($Ripk1^{K376R/+}Tnfr1^{−/−}$ mice: $n = 4$; $Ripk1^{K376R/−}Tnfr1^{−/−}$ mice: $n = 4$). **c** Immunohistochemical staining of cleaved Caspase3, CD11b, and F4/80 in skin sections of $Ripk1^{K376R/−}Tnfr1^{−/−}$ and $Ripk1^{K376R/+}Tnfr1^{−/−}$ littermate mice at P40 (scale bar, 50 μm). **d** Flow cytometry and statistical results of splenocytes stained with Ly6G and CD11b from $Ripk1^{K376R/−}Tnfr1^{−/−}$, $Ripk1^{K376R/−}Tnfr1^{+/−}$, and $Ripk1^{K376R/+}Tnfr1^{−/−}$ littermate mice at P40 ($Ripk1^{K376R/+}Tnfr1^{−/−}$ mice: $n = 4$; $Ripk1^{K376R/−}Tnfr1^{+/−}$ mice: $n = 4$; $Ripk1^{K376R/−}Tnfr1^{−/−}$ mice: $n = 4$). CD11b+Ly6G+ cells were identified as neutrophils. **e** Representative macroscopic images of $Ripk1^{K376R/−}Ripk3^{−/−}Caspase8^{−/−}$ and $Ripk1^{K376R/+}Ripk3^{−/−}Caspase8^{−/−}$ littermate mice at P40. **f** H&E staining of liver and skin sections of $Ripk1^{K376R/−}Ripk3^{−/−}Caspase8^{−/−}$ and $Ripk1^{K376R/+}Ripk3^{−/−}Caspase8^{−/−}$ littermate mice at P40 (scale bar, 50 μm), and microscopic quantification of the epidermal thickness from H&E results ($Ripk1^{K376R/+}Ripk3^{−/−}Caspase8^{−/−}$ mice: $n = 4$; $Ripk1^{K376R/−}Ripk3^{−/−}Caspase8^{−/−}$ mice: $n = 4$). **g** Flow cytometry and statistical results of splenocytes stained with Ly6G and CD11b from $Ripk1^{K376R/−}Ripk3^{−/−}Caspase8^{−/−}$ and $Ripk1^{K376R/+}Ripk3^{−/−}Caspase8^{−/−}$ littermate mice at P40 ($Ripk1^{K376R/+}Ripk3^{−/−}Caspase8^{−/−}$ mice: $n = 4$; $Ripk1^{K376R/−}Ripk3^{−/−}Caspase8^{−/−}$ mice: $n = 4$). CD11b+Ly6G+ cells were identified as neutrophils. In **b**, **d**, **f**, **g**, data are mean ± s.e.m. Statistical significance was determined using a two-tailed unpaired t test, n.s., $P > 0.05$, **$P < 0.01$, ***$P < 0.001$ and ****$P < 0.0001$

TRADD (sc-7868), FADD (sc-6036), and p65 (sc-109) were purchased from Santa Cruz Biotechnology; antibodies against phospho-RIPK3 (ab222320), phospho-MLKL (ab196436), and MLKL (ab67942) were purchased from Abcam; antibodies against β-tubulin (BE0025-100), secondary horseradish peroxidase (HRP)-conjugated anti-rabbit (BE0108-100), or anti-mouse antibodies (BE0107-100) were obtained from Easy Bio; antibody against Flag (M20008) was purchased from Abmart; recombinant mouse TNFα was purchased from R&D (410-MT-010). K63-TUBE (tandem ubiquitin-binding entities) (UM604) and M1-TUBE(UM206) were purchased from LifeSensors. CHX (C-6255), necrostatin-1(N9037), IKK inhibitor (T1452) and TAK1 inhibitor (O9890) were purchased from Sigma. MK2 inhibitor (4279/10) was obtained from Tocris Bioscience. zVAD.fmk (C1202) was obtained from Beyotime. Smac-mimetic BV6 (S7597) was obtained from Selleckchem. Flag-mTNFα (membrane-associated) (ALX-522-009-C050) and TRAIL (BML-SE722-0100) was purchased from Enzo Life Science. eFluor-450-conjugated anti-mouse B220 (48-0452-80), APC-cy7-conjugated anti-mouse CD45.2 (47-0454-82), and V500-conjugated cell viability dye (65-0866-18) were purchased from eBioscience; FITC-conjugated anti-mouse Ly6G (551460), FITC-conjugated anti-mouse CD19 (553785), and PerCP-cyanine5.5-conjugated anti-mouse CD4 (560782) antibodies were from BD Bioscience; APC-conjugated anti-mouse CD11b (101211) and PE-conjugated anti-mouse CD8 (100734) antibodies were from BioLegend. Antibodies for western-blotting analysis were used with 500- or 1000-fold dilution and for flow cytometry were used with 200-fold dilution.

**Cell culture**. HEK293T (purchased from ATCC) and MEF cells were cultured in Dulbecco's modified Eagle's medium (DMEM) medium (Gibco) supplemented with 10% fetal bovine serum, non-essential amino acids, sodium pyruvate, penicillin, and streptomycin. All cells were cultured at 37 °C and 5% $CO_2$.

**Cell infection**. A lentiviral supernatant was collected 48 h after co-transfection of expression plasmids (lenti-iκBα-SR-BSD, lenti-SV40-T-BSD, lenti-TAK1-Flag-BSD, lenti-TAB1-Flag-BSD, lenti-TAK1△N-Flag-BSD, lenti-MK2-Flag-BSD, lenti-IKKβ-CA-Flag-BSD) with packaging plasmids (psPAX2 and pMD2.G) into HEK293T cells. Viral supernatants were collected after 48 h, and target cells were incubated with the supernatant in the presence of polybrene for 8–12 h. After infection with virus, viral supernatants were replaced with fresh medium. After 24 h, infected cells were selected using blasticidin. The infection efficiency was determined by using western-blotting analysis.

**Generation and immortalization of MEFs**. $Ripk1^{+/+}$, $Ripk1^{K376R/+}$, $Ripk1^{K376R/K376R}$, $Ripk1^{K376R/−}$, and $Ripk1^{−/−}$ primary MEFs were generated from E11.5 embryos. After removing the placenta, yolk sac, head, and the dark red organs, embryos were finely minced and digested for 20 min in 0.25% trypsin. Single-cell suspension was then obtained by pipetting up and down the digested embryos. At passages 3 to 5, primary MEFs were immortalized by infection with SV40T-expressing lentivirus.

**Quantitative real-time PCR**. Total RNA was isolated using TRIzol (Invitrogen) and reverse transcribed using SuperScript III (Invitrogen). Quantitative real-time PCR (qRT-PCR) was performed using Power SYBR Green PCR Master Mix (Genestar). The amounts of transcript were normalized to those for glyceraldehyde 3-phosphate dehydrogenase. Melting curves were run to ensure amplification of a single product. The primers used are listed in Supplementary Table 3.

**Analysis of cell death and Caspase3 activity**. MEFs were seeded the day before at 10,000 per well in duplicates in a 96-well plate. The next day, cells were pre-treated with the indicated compounds for 60 min and then stimulated with the indicated concentration of mTNFα in the presence of 5 μM SytoxGreen

(Invitrogen, S34860) or 20 μM Ac-DEVD-MCA (Enzo Life Science, ALX-260-031-M005). SytoxGreen intensity and Caspase3 activation were measured at intervals of 1 h using a Fluostar Omega fluorescence plate reader, with an excitation filter of 485 nm (SytoxGreen) or 360 nm (Ac-DEVD-MCA), an emission filter of 520 nm (SytoxGreen) or 460 nm (Ac-DEVD-MCA). The percentage of cell death was calculated as (induced fluorescence − background fluorescence)/(maximum fluorescence − background fluorescence) × 100. The maximal fluorescence is obtained by full permeabilization of the cells by using Triton X-100. All cell death and Caspase3 activation data were presented as mean ± s.e.m. of three independent experiments.

**Western-blotting, immunoprecipitation, and ubiquitination assays**. Cells were lysed in lysis buffer (50 mM HEPES, pH 7.4, 150 mM NaCl, 1% NP-40, 1 mM EDTA) containing 1 mM sodium orthovanadate, 1 mM sodium fluoride, 1 mM phenylmethylsulfonyl fluoride (PMSF), and a protease inhibitor mixture (Roche). Cell lysates were then subjected to sodium dodecyl sulfate-polyacrylamide gel electrophoresis (SDS-PAGE) and transferred onto a polyvinylidene difluoride membrane (Millipore). The membrane was sequentially probed with primary antibodies and HRP-conjugated secondary antibodies. ECL substrates (Pierce) were used to visualize the specific bands on the membrane. Original blots are provided in the Source Data file.

For Complex I and Complex II immunoprecipitation, treated cells in 15 cm dishes were washed with cold phosphate-buffered saline (PBS), and then lysed in lysis buffer (30 mM Tris-HCl, pH 7.4, 120 mM NaCl, 2 mM EDTA, 2 mM KCl, 10% glycerol, and 1% Triton X-100) containing 1 mM sodium orthovanadate, 1 mM sodium fluoride, 1 mM PMSF, and a protease inhibitor mixture (Roche). Cell lysates were rotated at 4 °C for 30 min, and then clarified at 4 °C. After centrifugation at $18,000 \times g$ for 10 min, protein lysis was immunoprecipitated with antibody-conjugated beads overnight. For Complex I, using anti-FLAG M2 beads (A2220; Sigma); for Complex II, using RIPK1 antibodies plus Protein A/G agarose. Beads were washed four times in lysis buffer and samples were eluted by boiling in 10 μl 5× SDS loading dye. The input and immunoprecipitated samples were then subjected to SDS-PAGE.

For ubiquitination assay, cells were lysed in RIPA lysis buffer (50 mM Tris, pH 7.4, 150 mM NaCl, 1 mM EDTA, 20 mM N-ethylmaleimide, and 1% Triton X-100). Lysates were immediately boiled for 10 min in the presence of 1% (vol/vol) SDS and then were diluted with a lysis buffer until the concentration of SDS was decreased to 0.1%. Immunoprecipitates were analyzed by immunoblot with anti-K63 ubiquitin antibody.

**TUBE purification**. Following specified stimulations, ice-cold PBS-washed MEFs ($1 \times 10^7$) were harvested in 500 μl of 6 M urea lysis buffer (100 mM Tris-HCl, pH 8.0, 0.15 M NaCl, 5 mM EDTA, 1.5 mM $MgCl_2$, 1% Triton X-100, Roche cOmplete Protease Inhibitor Cocktail, 1 mM NEM) and flash frozen in liquid nitrogen. Thawed lysates were centrifuged for 30 min at 4 °C and diluted to 1 ml using dilution buffer (without urea) to bring the urea to 3 M. K63-TUBE or M1-TUBE was then added to the lysates and then incubated at 4 °C for 2 h, followed by 3 h incubation with Flag resin. After 4× washes in washing buffer (100 mM Tris-HCl, pH 8.0, 0.15 M NaCl, 5 mM EDTA, 0.05% NP-40), bound proteins were eluted using reducing and denaturing western blot sample buffer.

**Flow cytometry analyses**. Lymphocytes were isolated from the spleen. Red blood cells were lysed using RBC lysis buffer. A single-cell suspension was used for staining cell surface markers following standard protocols. Gating strategies are shown in Supplementary Fig. 13, and data acquisition was performed using a FACSAria II cytometer (BD). Flow cytometric data were analyzed with the FlowJo software.

**Histology and immunofluorescence**. For histopathological analyses, skin and other organs were fixed in 4% paraformaldehyde solution, processed according to standard procedures, embedded in paraffin, and sectioned. Five-micrometer-thick sections were stained with H&E, immunohistochemistry, and immunofluorescence. We used the following antibodies: F4/80 (Ceville), Ly6G (Ceville), CD11c (Ceville), CD3 (Ceville), CD45 (Ceville), K10 (Abcam), K14 (Abcam), loricirin (Abcam), cleaved Caspase3 (CST), and cleaved Caspase8 (CST). TUNEL staining was performed using the Apoptag kit (EMD Millipore). ImageJ software was applied to evaluate the ratio of positive cells in the total regions.

**Measurement of serum ALT and AST**. Blood samples were centrifuged at room temperature and the supernatant serum was collected. Serum samples were frozen in liquid nitrogen and stored at −80 °C. Serum levels of ALT and AST were measured by using microplate test kits (#C010-2) purchased from Nanjing Jiancheng Bioengineering Institute.

**Cytokine measurement**. Cytokines in serum or organ homogenates were detected by Ready-SET-GO ELISA kits (eBioscience). All samples were measured in triplicate according to the manufacturer's protocol.

**Statistics**. All values in this article were given as mean ± s.e.m., unless stated otherwise. All experiments were reproduced at least three independent times, and results shown in this article were representative. Statistical significance was calculated by two-tailed unpaired $t$ test using GraphPad Prism software. Statistical significance was set based on $P$ values. n.s., $P > 0.05$; *$P < 0.05$; **$P < 0.01$; ***$P < 0.001$; ****$P < 0.0001$.

**Reporting summary**. Further information on research design is available in the Nature Research Reporting Summary linked to this article.

## Data availability
The authors declare that the data supporting the findings of this study are available within the paper and its Supplementary Information Files. The source data underlying Figs. 1e–g, 2a–h, 3a–h, 4c, g–i, 5b, e–g, 6c, f–i, 7b, d, f, g and Supplementary Figs. 2a–d, 3a–m, 4e, h–i, 5a–e, 6b, 7f–j, 8b, d, 9a, d–e, 10d–f, 11a, c, d are provided as a Source Data file.

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

## Acknowledgements

This work was partially supported by grants (31930039, 81570211 and 81630058 to X.L., and 31670904 to X.Z.) from National Natural Science Foundation of China, and funding from Tsinghua University-Beijing University Jointed Center for Life Sciences. We thank Yue Liu for helping to draw the cartoon model.

## Author contributions

Y.T., H.T., X.Z., and X.L. designed the experiments. Y.T. and H.T. performed most of the experiments. J.Z helped to perform some mouse experiments. Y.W. and J.Q. provided technical help and gave conceptual advice. Y.T., H.T., and X.L. wrote the manuscript.

## Additional information

**Competing interests:** The authors declare no competing interests.

**Peer Review Information** *Nature Communications* thanks the anonymous reviewers for their contribution to the peer review of this work. Peer reviewer reports are available.

