## [Peer Review File · Nature Communications]

Reviewers' comments:

Reviewer #1 (Remarks to the Author):

In this manuscript, Tang et al. describe knock-in mice with the ubiquitin acceptor lysine residue K376 in RIPK1 mutated to arginine. Homozygous Ripk1 K376R/K376R mice showed embryonic lethality associated with aberrant cell death primarily in the fetal liver. This lethality could be delayed by TNFR1 ablation, but was rescued by loss of the cell death mediators caspase 8 and RIPK3. Mechanistically the authors propose that K63-linked ubiquitination of RIPK1 on K376 restrains its pro-death function through the activation of TAK1. In a final part of the manuscript the authors characterize Ripk1 K376R/Ko mice, which in contrast to Ripk1 K376R/K376R were viable but suffered of systemic RIPK3-dependent inflammation. The authors link this difference in viability to a Ripk1 K376R gene dosage effect.

Overall, the study confirms the predicted and based on in vitro data expected importance of RIPK1 ubiquitination on K376 using mouse genetics. These in vivo results are novel and the rescue crosses are well done. However, there are issues with the molecular mechanism and the used Ripk1 Ko allele, which lacks a thorough characterization and might not be a functional null allele.

1. Is the RIPK1 kinase domain still expressed from the Ripk1 Ko allele (Fig. 1a)? The authors should show uncropped western blots using an antibody against the RIPK1 kinase domain. If this fragment is still expressed, could the isolated RIPK1 kinase domain function as a decoy substrate for RIPK1 or still interact with the RIPK3 kinase domain? The presented data do not exclude such possibilities. As a consequence, data in Fig. 6 and 7 cannot be clearly interpreted. Crosses to a validated Ripk1 null allele are absolutely necessary to make these conclusions. Based on their model Ripk1 K376R/Wt mice might also show a mild phenotype especially when challenged. Is this the case? Ripk1 K376R/Wt and Ripk1 Ko/Ko cells should be included in the analysis in Fig. 6 g to i.
2. The molecular mechanism of how ubiquitination of RIPK1 restrains its cell death function is not well enunciated. As the authors correctly state RIPK1 can be subject to various ubiquitination events including K63, K48, K11 and M1-linked chains. The authors should therefore analyze all these linkages. Preferentially, IPs should be performed under denaturing conditions to assay solely RIPK1 ubiquitination. Is linear or K63 ubiquitination of complex I in general affected? Fig. 3b curiously shows increased RIPK1 modification within complex I in Ripk1 K376R/K376R cells, which the authors claim is phosphorylated RIPK1 (line 147 and 161). This needs to be confirmed by p-RIPK1 S166 western blotting and DUB/phosphatase treatment. The authors claim that TAK1 kinase activity is reduced in Ripk1 K376R/K376R cells (line 177), however data to support this statement is missing. The authors should validate this by assaying TAK1 and downstream MAPK activation (ERK, JNK, p38, MK2) and RIPK1 phosphorylation on inhibitory sites.

Specific points:

- Fig. 1b, S2a and S4a: Please indicate animal age.
- Fig. 1c: Embryo numbers and genotypes at different embryonic stages should be shown.
- Fig. 1e and f: Quantifications of microscopy data are missing.
- Fig. 2 and b: How does RIPK1 and Nec1 regulated TC-induced apoptosis? This form of cell death is thought to be RIPK1 independent.
- Fig. 2: Cell death in response to TNF alone should be analyzed.
- Does Nec1 inhibit caspase activation in Ripk1 K376R/K376R cells? In this context it is interesting to note that RIPK1 kinase activity may contribute to the induction of both apoptosis and necroptosis under certain conditions.
- Fig. 2d: MLKL activation should be analyzed.
- Fig. 2e: p-RIPK3 levels look strange. RIPK1 levels are missing.
- Line 158: Data are missing in Fig. 3d to show reduced NF-kB activation. Cells lacking IκBα-SR expression should be analyzed in the same experiment for direct comparison. Cell death data should be included in addition to western blotting for caspase activation.
- Absolute cell numbers should be shown for all FACS experiments.
- Fig. 5: Does RIPK3 loss lead to a partial rescue?
- Fig. S3b: Total MLKL levels are missing.
- Lines 72 and 300: TAK1 is thought to inhibit and not activate RIPK1.
- Lines 198 and 211: The authors claim that the Ripk1 K376R mutation affects skin and immune cell development. Differences are caused by cell death and are not developmental defects. This needs to be clarified.
- RIPK3 is thought to have function in addition to driving necroptosis. For example, lines 24 and 292 therefore need to be corrected.
- The lethal phenotype of the Ripk1 K376R/K376R mice is reminiscent of RelA or NEMO Ko mice, which are rescued by kinase dead RIPK1 (Vlantis et al 2016), but this is not mentioned in the discussion. Nor do the authors discuss why Ripk1 K376R appears to selectively promote fetal liver cell death, whereas the endothelial cells that undergo RIPK1-dependent cell death in other mouse models are spared. A discussion of the TNFR1-independent forms of cell death in Ripk1 K376R/K376R mice is missing as well.

Reviewer #2 (Remarks to the Author):

Review 18th December 2018

- You should be trying to help the work get published not necessarily in this journal but ultimately.
- Don't criticize an experiment unless you can tell the authors how they could do it better. "If you just want to throw darts," he would say, "go to the pub."
- Keep in mind that no one ever built a statue to a critic.
- Try to act as a peer in the process of peer review.

Science Signaling 2009 Michael Yaffe

Title: K63-linked ubiquitination regulates RIPK1 kinase activity to prevent cell death during embryogenesis and inflammation

Manuscript # NCOMMS1834785

General Remarks

This is an interesting and important piece of work further extending our understanding of the role of RIPK1 in regulating inflammation. I think it will be of interest to the field and Nat Comms is an appropriate venue, even though I found it a bit confusing in places (the heterozygotes) and I think there is room for improvement. It shows that Ripk1K376A/K376A mice and Ripk1^{-/-} mice are not equivalent with the former dying at ~E11. Figures 2 & 3 have some biochemical analysis of Ripk1K376A/K376A cells. Figure 4 shows that the early embryonic lethality is driven by TNFR1 and Figure 5 by RIPK3 and caspase-8 mediated cell death. Figure 6 and 7 look at the heterozygote phenotypes from which one can at least conclude that there is a dose response.

In my opinion this manuscript misses a real opportunity to provide some light into the function of the K377 ubiquitination. Figure 4 is an example, the skin phenotype and role of TNFR1 were looked

at in RIPK1^{-/-} mice in Rickard et al. They showed that the RIPK1 knock-out had inflamed skin and that was not affected by loss of TNFR1. The authors can't analyse the phenotype of the K377 homozygote mice because they die too early. However they could directly compare RIPK1 TNFR1 double knock-out mice. They could look at the skin, intestine and cytokines in these mice. The overriding question is to what extent the K376 mutant behaves like a RIPK1^{-/-} mouse. Because of the slight difference in onset and severity of RIPK1^{-/-} phenotype in Rickard et al I think these authors need to analyse these mice themselves so that readers can directly compare phenotypes. But the same is true in Figure 3, how much is due to loss of RIPK1 and how much is due to loss of K376? I believe the manuscript will be much stronger if some of the assays in Figure 3 are done with Ripk1^{-/-} cells as reference.

Specific Remarks

1. Fig. 1f is rather unimpressive, need to have appropriate controls of Casp8^{-/-} mice (can be on a RIPK3^{-/-} background).
2. Fig. 2e massive phospho-RIPK3 activity in absence of any cell death stimulus, which does not increase in response to TNF chx why? Yet in Fig. 2f very different picture.
3. Big problem with figure 2 is that treatment with TNF alone not done. Important to understand the physiological role of K376.
4. Fig. 2f how is RIPK3 activated in the absence of RIPK1? Given the rather strange results I recommend doing the cell death assays with the RIPK1 knock-out cells. According to He et al Cell 2009 RIPK1 should be required for TNF induced necroptosis. The only known exception to this that I know is from "Moujalled, et al (2013). Cell Death & Disease, 4(1), e465-. doi:10.1038/cddis.2012.201". But this was a situation where RIPK3 was over expressed.
5. Fig. 3a very unconvincing. RIPK1 ladder is normally very distinctive. don't believe that it shows, line 143 "Ripk1K376R/K376R mutation significantly reduced the K63-linked ubiquitination"
6. Fig. 3c NF-κB activation still occurs and reduction is only minor. This is really different to the standard models one sees where ubiquitylation at K376 provides the scaffold for kinase activation and where one would predict that activation should be completely lost. I think this should be brought out more in discussion etc.
7. line 161, the only conclusion that can be made is "not solely due"
8. line 185 - Also reference 29 & 30
9. line 188 - weaning typically occurs 3-4 weeks after birth. If the mice die at day 12 they cannot be "weaned at Mendelian ratios". Fig. S2A makes no reference to when these numbers are relevant - at weaning? or earlier?

Reviewer #3 (Remarks to the Author):

This manuscript reports how RIPK1 K376 ubiquitination is required for optimal TNF-induced NF κ B, and also for preventing TNF-killing, likely by inhibiting lethal RIPK1 kinase activity. Notably, the authors have generated a RIPK1 K376R mouse, to genetically demonstrate for the first time, the critical role for RIPK1 K376 ubiquitination for animal viability. Overall, the rigorous genetic experiments are well performed and the conclusions drawn from these are sound. The major area that the manuscript could be significantly improved is by a more thorough mechanistic analysis of how RIPK1 K376 impacts TNF signalling, as highlighted in the specific comments below:

1. Some mention of other ubiquitin ligases implicated in TNF death decisions, including RIPK1 regulation, is warranted (e.g. <https://doi.org/10.1016/j.celrep.2018.03.054>; <http://dx.doi.org/10.1016/j.celrep.2016.04.032>).
2. Fig 1, Fig 2. etc. Please define error bars clearly. i.e. Fig. 1g/2 represent combined results from 3 independent mice, 3 experimental replicates etc.?
3. Fig. 2. The authors show relatively short time courses (5-6 h) to demonstrate how RIPK1 K376R MEFs are sensitized to TNF/CHX killing. It would be informative if the authors measured killing over a longer time-frame, particularly with TNF treatment alone (not just western blots as currently depicted in Fig. 2d), to assess the importance of K376 for survival, which is likely to better reflect the physiological responses (i.e. death/inflammation) observed in the RIPK1 K376R embryos. It would be interesting if a similar assessment could be undertaken for FasL/TRAIL stimulations to define how widespread RIPK1 K376 modification is for limiting death receptor killing, particularly as TNFR1 deficiency does not fully rescue the RIPK1 K376R phenotype, including post-natal RIPK3/caspase-8-driven lethality.
4. It has been documented how deletion of RIPK1 sensitizes to TNF and CHX killing (e.g. [doi:10.1038/cdd.2009.178](https://doi.org/10.1038/cdd.2009.178)), likely via the loss of TRAF2 and cIAP1 (DOI 10.1074/jbc.M110.216226). Therefore, it would be pertinent to examine if TRAF2/cIAP levels are depleted in RIPK1 K376R cells, and if this thereby might contribute to RIPK1 K376R death.
5. Fig 2D/2F. RIPK1 K376R appears less stable compared to WT RIPK1 upon TNF stimulation, and this is not blocked by the inhibition of caspase activity. Is this degradation blocked by proteasome inhibition? If so, could this suggest that RIPK1 K376 ubiquitylation acts to stabilize RIPK1, which might also prevent TRAF2/cIAP1 loss?
6. Fig 3A. This experiment is important and needs to be improved. First, RIPK1 KO cells should be used as a control to demonstrate specificity. Second, this experiment does not prove that the ubiquitin chains detected are occurring on RIPK1. These chains could be on proteins that are interacting with RIPK1 complexes. Therefore, RIPK1 complexes should be dissociated, such as via the use of SDS, before diluting out the SDS and purifying RIPK1 in the absence of proteins that may also be ubiquitinated. Third, in Fig. 3A the amount (and input) of RIPK1 K376R purified is less than WT RIPK1, which makes it difficult to interpret the real reduction in K63-linked Ub chains in RIPK1 K376R cells. Finally, the authors should examine how total ubiquitinated RIPK1 (such as using the Tandem Ubiquitin Binding Entities (TUBE) from Life Sensors, or UBA-GST purification strategies) varies in WT

and RIPK1 K376R cells (using RIPK1 KO cells as controls) following TNF stimulation over time. Similarly, because RIPK1 K376R appears unstable following TNF stimulation, which may reflect proteasomal degradation, some analysis of how RIPK1 modification is impacted, or not, in RIPK1 K376R cells, with other ubiquitin linkages, such as K48, is warranted, and would be of significant interest to define.

7. Fig. 3C. It would be useful to have included RIPK1 KO cells in this experiment, as it has been reported that RIPK1 is dispensable to TNF-induced NF κ B responses in MEFs (i.e. doi: 10.1038/cdd.2009.178). If the authors RIPK1 KO MEFs show no defect in NF κ B, then this will alter the conclusion as to how RIPK1 K376R impacts the TNF signalling response.

8. Fig.3D. The effectiveness of the I κ B α SR in WT and RIPK1 K376R cells needs to be compared/documentated in order to conclude that increased RIPK1 K376R death is not, at least in part, a consequence of defective NF κ B.

9. Fig3E and 3F. RIPK1 K367 purification is significantly higher than WT cells. This complicates the interpretation of the IPs in Figs 3E and 3F and the authors conclusions that there is increased association of caspase-8 and cFLIP, because as more RIPK1 K376R is pulled down, it would be expected that more caspase-8 and cFLIP will be co-purified.

10. Figure 2 (cell death analysis) and western blots in Figure 3. In general, it would be beneficial to use RIPK1 KO cells in these experiments to allow i) a comparison on how RIPK1 K376R behaves, or does not behave, differently to RIPK1 deletion and, ii) help control for RIPK1 purification and antibody specificity.

11. The authors conclude that the experiments in Figure 3 demonstrate that RIPK1 K376 ubiquitination limits RIPK1 kinase activity by "...inhibiting TAK1 activity". Is it clearer to say that the loss of RIPK1 K376 ubiquitination likely prevents TAK1 recruitment into the RIPK1 complex, and thereby TAK1/p38/MK2 cannot target RIPK1 on S321/336 to prevent TNF killing? More experiments should be performed to test this idea; is TAK1/p38 recruitment, like the reduced MK2 recruitment shown in Fig. 3B, diminished in RIPK1 K376R cells? Is MK2-mediated RIPK1 phosphorylation blocked in RIPK1 K376R cells? Does M2 or TAK1 or IKK inhibition further sensitize RIPK1 K376R cells to TNF killing?

First, we would like to thank reviewers for their constructive comments and critiques. Based on their comments and critiques, we have performed a series of experiments. As results, we have added total 42 new panels as Fig. 2a, 2b, 2c, 2d, 2f, 2h, 3a, 3c, 3d, 3e, 3f, 3h, 6g. Supplementary Fig. 1a, 2a, 2b, 2c, 2d, 3b, 3d, 3e, 3h, 3i, 3j, 3k, 3l, 3m, 4d, 4e, 4f, 4g, 5c, 5d, 5e, 7i, 7j, 8a, 8b, 8c, 8d, 8e, 8f. We also revised Fig. 1e, 1f, 3b, 4i, 5g, 6f, 7d, 7g, Supplementary Fig. 4i, 5b, 6b, 7g, 9e, 10f, 11d. In addition, we provide the following point-to-point response to address reviewers' comments and critiques.

Reviewers' comments:

Reviewer #1 (Remarks to the Author):

In this manuscript, Tang et al. describe knock-in mice with the ubiquitin acceptor lysine residue K376 in RIPK1 mutated to arginine. Homozygous Ripk1 K376R/K376R mice showed embryonic lethality associated with aberrant cell death primarily in the fetal liver. This lethality could be delayed by TNFR1 ablation, but was rescued by loss of the cell death mediators caspase 8 and RIPK3. Mechanistically the authors propose that K63-linked ubiquitination of RIPK1 on K376 restrains its pro-death function through the activation of TAK1. In a final part of the manuscript the authors characterize Ripk1 K376R/Ko mice, which in contrast to Ripk1 K376R/K376R were viable but suffered of systemic RIPK3-dependent inflammation. The authors link this difference in viability to a Ripk1 K376R gene dosage effect.

Overall, the study confirms the predicted and based on in vitro data expected importance of RIPK1 ubiquitination on K376 using mouse genetics. These in vivo results are novel and the rescue crosses are well done. However, there are issues with the molecular mechanism and the used Ripk1 Ko allele, which lacks a thorough characterization and might not be a functional null allele.

1. (1) Is the RIPK1 kinase domain still expressed from the Ripk1 Ko allele (Fig. 1a)? The authors should show uncropped western blots using an antibody against the RIPK1 kinase domain. If this fragment is still expressed, could the isolated RIPK1 kinase domain function as a decoy substrate for RIPK1 or still interact with the RIPK3 kinase domain? The presented data do not exclude such possibilities. (2) As a consequence, data in Fig. 6 and 7 cannot be clearly interpreted. Crosses to a validated Ripk1 null allele are absolutely necessary to make these conclusions. (3) Based on their model Ripk1 K376R/Wt mice might also show a mild phenotype especially when challenged. Is this the case? Ripk1 K376R/Wt and Ripk1 Ko/Ko cells should be included in the analysis in Fig. 6 g to i.

Response:

(1) Yes, the truncated RIPK1 has a pretty low expression in Ripk1 KO cells (Supple Fig.2a). We can detect this truncated band after long exposure. However, by co-immunoprecipitation assay, we can not detect the interaction between truncated RIPK1 with RIPK3, while full length RIPK1 interacts with RIPK3 (Supple Fig.2b), indicating that truncated RIPK1 may not be functional.

(2) This is a good concern. To rule out this possibility, we also generated another RIPK1 knockout mouse strain (*KD-Ripk1*^{-/-}), in which the translation of RIPK1 stops at Val42 and does not express

a short fragment (Supple Fig.8a,8b). As requested by this reviewer, we crossed the *KD-Ripk1*^{-/-} mice with *Ripk1*^{K376R/+} mice, and the *KD-Ripk1*^{K376R/-} mice showed the same phenotype as *Ripk1*^{K376R/-} mice (Supple Fig.8). These data further supported our conclusion.

(3) We cannot observe obvious inflammation in *Ripk1*^{K376R/+} mice. The *Ripk1*^{K376R/+} cells also have the same phenotype as WT cells after stimulation (Fig.6g and supple Fig.7i-7j). We added the *Ripk1*^{K376R/+} cells and *Ripk1*^{-/-} cells in the analysis in Fig.6g and supple Fig.7i-7j.

2. The molecular mechanism of how ubiquitination of RIPK1 restrains its cell death function is not well enunciated. (1) As the authors correctly state RIPK1 can be subject to various ubiquitination events including K63, K48, K11 and M1-linked chains. The authors should therefore analyze all these linkages. Preferentially, IPs should be performed under denaturing conditions to assay solely RIPK1 ubiquitination. Is linear or K63 ubiquitination of complex I in general affected? (2) Fig. 3b curiously shows increased RIPK1 modification within complex I in *Ripk1* K376R/K376R cells, which the authors claim is phosphorylated RIPK1 (line 147 and 161). This needs to be confirmed by p-RIPK1 S166 western blotting and DUB/phosphatase treatment. (3) The authors claim that TAK1 kinase activity is reduced in *Ripk1* K376R/K376R cells (line 177), however data to support this statement is missing. The authors should validate this by assaying TAK1 and downstream MAPK activation (ERK, JNK, p38, MK2) and RIPK1 phosphorylation on inhibitory sites.

Response:

(1) Yes, all the ubiquitination related IPs were performed under denaturing conditions. We also tried to analyze all ubiquitinated linkages on RIPK1, but some antibodies did not work well. To further confirm K376R is a K63-linked ubiquitinated site, we used the Tandem Ubiquitin Binding Entities (TUBE) assays for detecting K63-Ubi and M1-Ubi on RIPK1. From the results (Fig.3a and Supplementary Fig.3b), we can observe that K376R mutation could significantly decrease K63-Ubi of RIPK1 (Fig.3a), but had no effect on M1-Ubi of RIPK1(Supplementary Fig.3b).

We also tried to detect all ubiquitinated linkages in complex I, but we could only detect M1-Ubi and total-Ubi in complex I, K63-Ubi cannot be detected (maybe the antibody did not work well or the dose of K63-Ubi was too low in complex I). From the results below (see attached data below), M1-Ubi and total-Ubi of complex I were not affected by K376R mutation. As a control, RIPK1 deficiency significantly reduced M1-Ubi in complex I but had no effect on total-Ubi. Since complex I has a lot of ubiquitinated proteins, it is hard to clearly elucidate this result. It is also difficult to determine the effect on ubiquitinated events in complex I by decreased K63-linked ubiquitination of RIPK1. Nevertheless, we showed that K63-linked ubiquitination of RIPK1 was significantly reduced in *RIPK1*^{K376R/K376R} cells directly (Fig. 3a).

(2) We confirmed this result by p-RIPK1 antibody and showed the increased pRIPK1 in *Ripk1*^{K376R/K376R} cells (Fig.3b).

(3) We detected the downstream MAPK activation of TAK1 (Supple Fig.3m), and the results showed that phosphorylation of JNK, p38, MK2 were significantly inhibited in *Ripk1*^{K376R/K376R} cells (phosphorylated ERK has no significant difference). This is consistent with our result that K63-linked ubiquitination on K376R reduced the recruitment of TAK1 (Fig. 3b), thereby reducing TAK1 kinase-dependent activation of downstream kinases. Since there is no commercial antibody for RIPK1 phosphorylation site by TAK1, we could not directly detect TAK1-dependent phosphorylation of RIPK1.

Specific points:

- Fig. 1b, S2a and S4a: Please indicate animal age.

Response: Thank for this comment. We added the animal age in figure legend.

- Fig. 1c: Embryo numbers and genotypes at different embryonic stages should be shown.

Response: We have a statistic data in Supplementary Fig.1a.

- Fig. 1e and f: Quantifications of microscopy data are missing.

Response: We added the quantifications of microscopic data.

- Fig. 2 and b: How does RIPK1 and Nec1 regulated TC-induced apoptosis? This form of cell death is thought to be RIPK1-independent.

Response: In general, Nec-1 can inhibit necroptosis of WT MEFs induced by TNF α /CHX/zVAD but not apoptosis induced by TNF α /CHX. However, in *Ripk1*^{K376R/K376R} MEFs, Nec-1 can inhibit cell death induced by TNF α /CHX. The similar phenomenon has also been found in ABIN-1 KO MEFs (DOI: 10.1038/s41556-017-0003-1, please see figure3 in the article). It remains unclear about the mechanism for this phenomenon.

- Fig. 2: Cell death in response to TNF alone should be analyzed.

Response: We analyzed TNF-induced cell death and added the results in Fig.2a and Supplementary Fig.2c.

- Does Nec1 inhibit caspase activation in Ripk1 K376R/K376R cells? In this context it is interesting to note that RIPK1 kinase activity may contribute to the induction of both apoptosis and necroptosis under certain conditions.

Response: Yes, RIPK1 kinase activity can regulate both apoptosis and necroptosis. From our results, Nec-1 can inhibit caspase activation in *Ripk1*^{K376R/K376R} cells under TNFa/Smac or TNFa/CHX stimulation (Fig.2h and Supplementary Fig.2d).

- Fig. 2d: MLKL activation should be analyzed.

Response: We added MLKL activation in Fig.2d, but we cannot detect phosphorylated MLKL under TNFa stimulation alone.

- Fig. 2e: p-RIPK3 levels look strange. RIPK1 levels are missing.

Response: Sorry, this is a missed label. P-RIPK3 should be corrected to RIPK3. We have corrected Figure 2e.

- Line 158: Data are missing in Fig. 3d to show reduced NF-kB activation. Cells lacking IkBa-SR expression should be analyzed in the same experiment for direct comparison. Cell death data should be included in addition to western blotting for caspase activation.

Response: We added the cells lacking IkBa-SR expression and detect NF-kB activation and cell death (Fig.3d and Supple Fig. 3d-3e). IkBa-SR indeed inhibited NF-kB activation but cannot eliminate the difference of cell death in WT and *Ripk1*^{K376R/K376R} cells.

- Absolute cell numbers should be shown for all FACS experiments.

Response: We added the cell numbers for all FACS experiments.

- Fig. 5: Does RIPK3 loss lead to a partial rescue?

Response: From our observation, RIPK3 cannot rescue the lethality of *Ripk1*^{K376R/K376R} mice (Supple Fig.6c). We cannot rule out the possibility that RIPK3 can delay the embryonic lethality of *Ripk1*^{K376R/K376R} mice.

- Fig. S3b: Total MLKL levels are missing.

Response: We added the total MLKL level (now as Supple Fig.5b).

- Lines 72 and 300: TAK1 is thought to inhibit and not activate RIPK1.

Response: Thank you. We corrected the statement.

- Lines 198 and 211: The authors claim that the Ripk1 K376R mutation affects skin and immune cell development. Differences are caused by cell death and are not developmental defects. This needs to be clarified.

Response: Thank you. We corrected the statement.

- RIPK3 is thought to have function in addition to driving necroptosis. For example, lines 24 and 292 therefore need to be corrected.

Response: Thank you. We corrected the statement.

- The lethal phenotype of the Ripk1 K376R/K376R mice is reminiscent of RelA or NEMO Ko mice, which are rescued by kinase dead RIPK1 (Vlantis et al 2016), but this is not mentioned in the discussion. Nor do the authors discuss why Ripk1 K376R appears to selectively promote fetal liver cell death, whereas the endothelial cells that undergo RIPK1-dependent cell death in other mouse models are spared. A discussion of the TNFR1-independent forms of cell death in Ripk1 K376R/K376R mice is missing as well.

Response: Thank you. We added these discussions.

Reviewer #2 (Remarks to the Author):

General Remarks

This is an interesting and important piece of work further extending our understanding of the role of RIPK1 in regulating inflammation. I think it will be of interest to the field and Nat Comms is an appropriate venue, even though I found it a bit confusing in places (the heterozygotes) and I think there is room for improvement. It shows that Ripk1K376A/K376A mice and Ripk1^{-/-} mice are not equivalent with the former dying at ~E11. Figures 2 & 3 have some biochemical analysis of Ripk1K376A/K376A cells. Figure 4 shows that the early embryonic lethality is driven by TNFR1 and Figure 5 by RIPK3 and caspase-8 mediated cell death. Figure 6 and 7 look at the heterozygote phenotypes from which one can at least conclude that there is a dose response.

In my opinion this manuscript misses a real opportunity to provide some light into the function of the K377 ubiquitination. Figure 4 is an example, the skin phenotype and role of TNFR1 were looked at in RIPK1^{-/-} mice in Rickard et al. They showed that the RIPK1 knock-out had inflamed skin and that was not affected by loss of TNFR1. The authors can't analyze the phenotype of the K377 homozygote mice because they die too early. However, they could directly compare RIPK1 TNFR1 double knock-out mice. They could look at the skin, intestine and cytokines in these mice. The overriding question is to what extent the K376 mutant behaves like a RIPK1^{-/-} mouse. Because of the slight difference in onset and severity of RIPK1^{-/-} phenotype in Rickard et al I think these authors need to analyze these mice themselves so that readers can directly compare phenotypes. But the same is true in Figure 3, how much is due to loss of RIPK1 and how much is due to loss of K376? I believe the manuscript will be much stronger if some of the assays in Figure 3 are done with Ripk1^{-/-} cells as reference.

Response: Thanks for your suggestions to improve our manuscript. We compared *Ripk1^{-/-}Tnfr1^{-/-}* mice with *Ripk1^{K376R/K376R}Tnfr1^{-/-}* mice, and they displayed the similar phenotype (Supplementary Fig.4d-4g). The phenotype of our *Ripk1^{-/-}Tnfr1^{-/-}* mice were similar to another paper (DOI: <https://doi.org/10.1016/j.cell.2014.04.018>), but different from Rickard's paper.

We also used *Ripk1*^{-/-} KO cells as control in Figure 3.

Specific Remarks

1. Fig. 1f is rather unimpressive, need to have appropriate controls of Casp8^{-/-} mice (can be on a RIPK3^{-/-} background).

Response: We added *Ripk3/caspase8* dKO mice as control, since Casp8^{-/-} mice are lethal.

2. Fig. 2e massive phospho-RIPK3 activity in absence of any cell death stimulus, which does not increase in response to TNF chx why? Yet in Fig. 2f very different picture.

Response: Sorry, that was a missed labeling. P-RIPK3 should be corrected to RIPK3.

3. Big problem with figure 2 is that treatment with TNF alone not done. Important to understand the physiological role of K376.

Response: We added TNF alone stimulation. (Fig.2a and Supple. Fig 2c)

4. Fig. 2f how is RIPK3 activated in the absence of RIPK1? Given the rather strange results I recommend doing the cell death assays with the RIPK1 knock-out cells. According to He et al Cell 2009 RIPK1 should be required for TNF induced necroptosis. The only known exception to this that I know is from "Moujalled, et al (2013). Cell Death & Disease, 4(1), e465–. doi:10.1038/cddis.2012.201". But this was a situation where RIPK3 was over expressed.

Response: Thank you for this question. In general, necroptosis should be a RIPK1-dependent process. However, a recent study found RIPK1-deficient dendritic cells are more sensitive to necroptosis (doi:10.4049/jimmunol.1701229), indicating a RIPK1-independent necroptotic process.

We found that different stage MEFs of *Ripk1*^{-/-} (from embryos at E11.5 or E15.5) showed different necroptotic response to TNFa/zVAD stimulation, but similar apoptotic response. E15.5 MEFs are more sensitive to TNFa/zVAD-induced necroptosis, but E11.5 MEFs are resistant to TNFa/zVAD-induced necroptosis. The exact reason remains unclear. To be consistent with *Ripk1*^{K376R/K376R} MEFs (from embryos at E11.5), we used *Ripk1*-KO MEFs (from embryos at E11.5) instead to do the analysis. We also confirmed this result by using another set of *Ripk1*-KO MEFs from *KD-Ripk1*^{-/-} mice (data not shown). We did the cell death assays with *Ripk1*-KO cells (Fig.2a,2b).

5. Fig. 3a very unconvincing. RIPK1 ladder is normally very distinctive. don't believe that it shows, line 143 "Ripk1K376R/K376R mutation significantly reduced the K63-linked ubiquitination"

Response: We repeated this experiment with Tandem Ubiquitin Binding Entities (TUBE) assays for K63-Ub and M1-Ub. The results clearly showed *Ripk1*^{K376R/K376R} mutation significantly reduced the K63-Ub but not M1-Ub (Figure 3a and Supple.Fig3b).

6. Fig. 3c NF-κB activation still occurs and reduction is only minor. This is really different to the standard models one sees where ubiquitylation at K376 provides the scaffold for kinase activation

and where one would predict that activation should be completely lost. I think this should be brought out more in discussion etc.

Response: We have added RIPK1^{-/-} cells as a negative control in Fig. 3c, and showed that TNF α -induced nuclear translocation of p65 was completely defected in RIPK1^{-/-} cells, but it was slightly reduced in RIPK1^{K376R/K376R} cells (Fig. 3c). We have modified our discussion too.

7. line 161, the only conclusion that can be made is “not solely due”

Response: Thank you. We correct the statement.

8. line 185 - Also reference 29 & 30

Response: Thank you. We add the reference 29 & 30.

9. line 188 - weaning typically occurs 3-4 weeks after birth. If the mice die at day 12 they cannot be “weaned at Mendelian ratios”. Fig. S2A makes no reference to when these numbers are relevant - at weaning? or earlier?

Response: Thank you. We correct the statement and also add the mice age in Figure legend.

Reviewer #3 (Remarks to the Author):

This manuscript reports how RIPK1 K376 ubiquitination is required for optimal TNF-induced NF-kB, and also for preventing TNF-killing, likely by inhibiting lethal RIPK1 kinase activity. Notably, the authors have generated a RIPK1 K376R mouse, to genetically demonstrate for the first time, the critical role for RIPK1 K376 ubiquitination for animal viability. Overall, the rigorous genetic experiments are well performed and the conclusions drawn from these are sound. The major area that the manuscript could be significantly improved is by a more thorough mechanistic analysis of how RIPK1 K376 impacts TNF signaling, as highlighted in the specific comments below:

1. Some mention of other ubiquitin ligases implicated in TNF death decisions, including RIPK1 regulation, is warranted

(e.g. <https://doi.org/10.1016/j.celrep.2018.03.054>; <http://dx.doi.org/10.1016/j.celrep.2016.04.032>).

Response: We add these two references into the introduction.

2. Fig 1, Fig 2. etc. Please define error bars clearly. i.e. Fig. 1g/2 represent combined results from 3 independent mice, 3 experimental replicates etc.?

Response: We defined the error bars in figure legend.

3. Fig. 2. (1) The authors show relatively short time courses (5-6 h) to demonstrate how RIPK1 K376R MEFs are sensitized to TNF/CHX killing. It would be informative if the authors measured killing over a longer time-frame, particularly with TNF treatment alone (not just western blots as currently depicted in Fig. 2d), to assess the importance of K376 for survival, which is likely to better reflect the physiological responses (i.e. death/inflammation) observed in the RIPK1 K376R embryos. (2) It would be interesting if a similar assessment could be undertaken for FasL/TRAIL stimulations to define how widespread RIPK1 K376 modification is for limiting death receptor

killing, particularly as TNFR1 deficiency does not fully rescue the RIPK1 K376R phenotype, including post-natal RIPK3/caspase-8-driven lethality.

Response: (1) We did the cell death assay at different time points following for TNF stimulation alone (Fig.2a).

(2) We did these assays by for FasL/TRAIL stimulations (Supplementary Fig.5c-5e), and the results suggested RIPK1 K376 modification is also involved in the regulation of other death receptor induced cell death.

4. It has been documented how deletion of RIPK1 sensitizes to TNF and CHX killing (e.g. doi:10.1038/cdd.2009.178), likely via the loss of TRAF2 and cIAP1 (DOI 10.1074/jbc.M110.216226). Therefore, it would be pertinent to examine if TRAF2/cIAP levels are depleted in RIPK1 K376R cells, and if this thereby might contribute to RIPK1 K376R death.

Response: We detected expression level of TRAF2 and cIAP1 in *Ripk1*^{K376R/K376R} MEFs after stimulation, but they have no significant difference comparing to WT cells (Fig.2d/2f). As a control, expression of TRAF2/cIAP1 were decreased in *Ripk1*^{-/-} cells after stimulation. Therefore, expression level of TRAF2 and cIAP1 is not the reason for the increased cell death in *Ripk1*^{K376R/K376R} cells.

5. Fig 2D/2F. RIPK1 K376R appears less stable compared to WT RIPK1 upon TNF stimulation, and this is not blocked by the inhibition of caspase activity. Is this degradation blocked by proteasome inhibition? If so, could this suggest that RIPK1 K376 ubiquitylation acts to stabilize RIPK1, which might also prevent TRAF2/cIAP1 loss?

Response: Proteasome inhibitor MG132 cannot inhibit mutated RIPK1 degradation with TNF alone stimulation but can prevent mutated RIPK1 degradation with TNF/zVAD stimulation (see attached data shown below a, b). This result is consistent with a recent finding that formation of necrosome activated K48-ubiquitin-dependent proteasomal degradation of RIPK1 (DOI 10.1038/s41419-018-0672-0). It is possible that RIPK1 K376 ubiquitylation acts to stabilize RIPK1 under necroptosis condition but not apoptosis. However, K376R mutation of RIPK1 did not affect TRAF2/cIAP1 expression (Fig.2d/2f). As a control, expression of TRAF2/cIAP1 were decreased in *Ripk1*^{-/-} cells after stimulation. Therefore, RIPK1 K376 ubiquitylation does not prevent TRAF2/cIAP1 loss to inhibit cell death. Moreover, *Ripk1*^{K376R/K376R} cells are more sensitive to cell death than *Ripk1*^{-/-} cells (Fig.2), and stability of RIPK1 cannot explain this phenomenon. Since MG132 could inhibit RIPK1 activation (see attached data shown below b), it would be interesting to further study the role of RIPK1 K376R stability in the regulation of RIPK1 activation.

6. Fig 3A. This experiment is important and needs to be improved. First, RIPK1 KO cells should be

used as a control to demonstrate specificity. Second, this experiment does not prove that the ubiquitin chains detected are occurring on RIPK1. These chains could be on proteins that are interacting with RIPK1 complexes. Therefore, RIPK1 complexes should be dissociated, such as via the use of SDS, before diluting out the SDS and purifying RIPK1 in the absence of proteins that may also be ubiquitinated. Third, in Fig. 3A the amount (and input) of RIPK1 K376R purified is less than WT RIPK1, which makes it difficult to interpret the real reduction in K63-linked Ub chains in RIPK1 K376R cells. Finally, the authors should examine how total ubiquitinated RIPK1 (such as using the Tandem Ubiquitin Binding Entities (TUBE) from Life Sensors, or UBA-GST purification strategies) varies in WT and RIPK1 K376R cells (using RIPK1 KO as controls) following TNF stimulation over time. Similarly, because RIPK1 K376R appears unstable following TNF stimulation, which may reflect proteasomal degradation, some analysis of how RIPK1 modification is impacted, or not, in RIPK1 K376R cells, with other ubiquitin linkages, such as K48, is warranted, and would be of significant interest to define.

Response: Thanks for your suggestions. All the ubiquitination related IPs were performed under denaturing conditions. We added the RIPK1 KO cells as control and perform TUBE assay to detect RIPK1 ubiquitination. The results showed that K376R mutation of RIPK1 reduce K63-Ubi on RIPK1 (Fig.3a) but has no effect on M1-Ubi of RIPK1(Supple Fig.3b) After TNFa/zVAD stimulation, *Ripk1*^{K376R/K376R} cells showed increased but weak level of K48-Ubi (data shown below c). It is possible that RIPK1 K376 ubiquitylation acts to stabilize RIPK1 by preventing K48-Ubi. However, *Ripk1*^{K376R/K376R} cells are more sensitive to cell death than *Ripk1*^{-/-} cells (Fig.2), and stability of RIPK1 cannot explain this phenomenon. From our data in Fig.3 and Supple Fig.3, we concluded that TAK1-mediated suppression of RIPK1 kinase activity was the mechanism.

7. Fig. 3C. It would be useful to have included RIPK1 KO cells in this experiment, as it has been reported that RIPK1 is dispensable to TNF-induced NF-κB responses in MEFs (i.e. doi: 10.1038/cdd.2009.178). If the authors RIPK1 KO MEFs show no defect in NF-κB, then this will alter the conclusion as to how RIPK1 K376R impacts the TNF signaling response.

Response: We added KO cells and found NF-κB activation was defective in KO cells, but partially reduced in RIPK1 K376R cells (Fig.3c). This result is consistent with previous finding that RIPK1

is indispensable for NF- κ B activation (doi:10.1038/nature20559, please see Extended Data Figure 4a).

8. Fig.3D. The effectiveness of the IkbSR in WT and RIPK1 K376R cells needs to be compared/documentated in order to conclude that increased RIPK1 K376R death is not, at least in part, a consequence of defective NF-kB.

Response: We detected NF-kB activation in IkbSR-expressed cells, and the results showed that IkbSR could inhibit NF-kB activation. (Supplementary Fig.3d)

9. Fig. 3E and 3F. RIPK1 K367 purification is significantly higher than WT cells. This complicates the interpretation of the IPs in Figs 3E and 3F and the authors conclusions that there is increased association of caspase-8 and cFLIP, because as more RIPK1 K376R is pulled down, it would be expected that more caspase-8 and cFLIP will be co-purified.

Response: We repeated these results, and showed that with comparable level of RIPK1, RIPK1 K376R mutation indeed increase the association with FADD/caspase8/cFLIP/RIP3 (Figure 3e and 3f).

10. Figure 2 (cell death analysis) and western blots in Figure 3. In general, it would be beneficial to use RIPK1 KO cells in these experiments to allow i) a comparison on how RIPK1 K376R behaves, or does not behave, differently to RIPK1 deletion and, ii) help control for RIPK1 purification and antibody specificity.

Response: We added RIPK1 KO cells as control in Figure 2 and Figure 3.

11. The authors conclude that the experiments in Figure 3 demonstrate that RIPK1 K376 ubiquitination limits RIPK1 kinase activity by "...inhibiting TAK1 activity". Is it clearer to say that the loss of RIPK1 K376 ubiquitination likely prevents TAK1 recruitment into the RIPK1 complex, and thereby TAK1/p38/MK2 cannot target RIPK1 on S321/336 to prevent TNF killing? More experiments should be performed to test this idea; (1) is TAK1/p38 recruitment, like the reduced MK2 recruitment shown in Fig. 3B, diminished in RIPK1 K376R cells? (2) Is MK2-mediated RIPK1 phosphorylation blocked in RIPK1 K376R cells? (3) Does M2 or TAK1 or IKK inhibition further sensitize RIPK1 K376R cells to TNF killing?

Response: Thank for these suggestions. We performed more experiments to support our conclusion. (1) After TNFa stimulation, the interaction between TAK1 and RIPK1 is decreased in *Ripk1*^{K376R/K376R} MEFs (Supple Fig.3l), and TAK1 recruitment is decreased in complex I of *Ripk1*^{K376R/K376R} MEFs (Fig.3b). (2) Since there is no commercial antibody for RIPK1 phosphorylation site by MK2 or IKK, we could not directly detect it. (3) We found IKK and TAK1 inhibition could fully diminish the difference of cell death in WT and *Ripk1*^{K376R/K376R} MEFs, but MK2 inhibition could not (Fig.3h and Supple Fig.3g-3i). Overexpressed activated TAK1 or IKK β , but not MK2, can diminish the difference of cell death in WT and *Ripk1*^{K376R/K376R} MEFs (Fig.3g and Supple Fig.3j-3k). Taken together, we concluded that K63-ubi of RIPK1 on K376 mainly suppress RIPK1 kinase activity through TAK1-IKK axis.

REVIEWERS' COMMENTS:

Reviewer #1 (Remarks to the Author):

The revised manuscript by Tang et al. has adequately addressed the main concerns raised in the original submission. By generating a novel RIPK1 KO allele the authors provide genetic data to support their previously presented conclusions. In addition, further biochemical data are provided to back their mechanistic model.

However, some minor comments still need to be addressed:

- The RIPK1 mobility shift in Figure 3b needs to be further analyzed by DUB and phosphatase treatment.
- P-TAK levels are missing in Supplementary Figure 3m.
- The authors might want to mention the recent data on TBK1/IKKe-mediated regulation of RIPK1 in the introduction.
- Line 92/93 should state at one month old and not at birth. Line 221 should be changed as well.

Reviewer #2 (Remarks to the Author):

Review 2nd June 2019

Title: K63-linked ubiquitination regulates RIPK1 kinase activity to prevent cell death during embryogenesis and inflammation

Manuscript # NCOMMS1834785

General Remarks

The authors have made a good job of responding to the issues that were raised.

I have some minor points that can be very readily dealt with.

Line 190 I would write, "Consistent with previous studies, loss of RIPK1 led to TNF induced loss of TRAF2 and cIAP1, however, K376 mutation of RIPK1 did not affect TRAF2 and cIAP1 stability (Fig. 2d,2f)." and cite this paper in addition to the one already cited, doi:10.1242/jcs.075770.

Line 180 "indicating that the increased cell death in Ripk1K376R/K376R MEFs is not ONLY due to its defective NF- κ B activation"

Line 170 "found that p65 translocation was significantly decreased in Ripk1K376R/K376RMEFs (Fig.3c)." Significance is a statistical measure so can't be used here and furthermore this doesn't look very strong (ie the lay usage of significance) I would simply say it was decreased.

Reviewer #3 (Remarks to the Author):

The authors have thoroughly addressed all my comments with new, or improved, data, and consequently have strengthened the manuscript considerably. Congratulations on a very nice story.

Reviewer #1 (Remarks to the Author):

The revised manuscript by Tang et al. has adequately addressed the main concerns raised in the original submission. By generating a novel RIPK1 KO allele the authors provide genetic data to support their previously presented conclusions. In addition, further biochemical data are provided to back their mechanistic model.

However, some minor comments still need to be addressed:

- The RIPK1 mobility shift in Figure 3b needs to be further analyzed by DUB and phosphatase treatment.

Response: Thanks for the reviewer's suggestions. According to the suggestions, we used DUB (USP2, Figure a) and phosphatase (λ PPase, Figure b) treatment. From the results, we can see USP2 treatment decreased the RIPK1 mobility shift, but increased normal RIPK1 in complex I. However, λ PPase treatment had no effect on RIPK1 mobility shift. These suggested the RIPK1 mobility shift in *Ripk1*^{K376R/K376R} cells are mainly ubiquitinated RIPK1. This was consistent with our results that the level of p-RIPK1 (S166) is very low (Figure 3b). We also found K376R mutation can promote K48-Ubi on RIPK1, and previous studies suggest that K63-Ubi also has a crosstalk with Met1-Ubi on the regulation of RIPK1 activation (<https://www.nature.com/articles/s41556-017-0003-1>). The nature of this ubiquitination will need to be further analyzed and to determine how K63-Ubi on K376 regulate other ubiquitinated chains on RIPK1. To make it more rigorous, we removed the sentence "Remarkably, phosphorylation of RIPK1 on S166 was significantly elevated in *Ripk1*^{K376R/K376R} MEFs (Fig.3b), suggesting that K63-linked ubiquitination of RIPK1 could suppress RIPK1 kinase activity."

- P-TAK levels are missing in Supplementary Figure 3m.

Response: Thank you for your suggestion. We tried several p-TAK1 antibodies (Abcam, ab109404; CST,9339; CST,4531; CST,4508), however, they all did not work in our hands. So, we cannot directly detect p-TAK1.

- The authors might want to mention the recent data on TBK1/IKKe-mediated regulation of RIPK1 in the introduction.

Response: Thank you. We add this recent finding in the introduction.

- Line 92/93 should state at one month old and not at birth. Line 221 should be changed as well.

Response: Thank you. We correct these statements.

Reviewer #2 (Remarks to the Author):

Review 2nd June 2019

Title: K63-linked ubiquitination regulates RIPK1 kinase activity to prevent cell death during embryogenesis and inflammation

Manuscript # NCOMMS1834785

General Remarks

The authors have made a good job of responding to the issues that were raised. I have some minor points that can be very readily dealt with. Line 190 I would write, "Consistent with previous studies, loss of RIPK1 led to TNF induced loss of TRAF2 and cIAP1, however, K376 mutation of RIPK1 did not affect TRAF2 and cIAP1 stability (Fig. 2d,2f)." and cite this paper in addition to the one already cited, doi:10.1242/jcs.075770. Line 180 "indicating that the increased cell death in Ripk1K376R/K376R MEFs is not ONLY due to its defective NF- κ B activation" Line 170 "found that p65 translocation was significantly decreased in Ripk1K376R/K376RMEFs (Fig.3c)." Significance is a statistical measure so can't be used here and furthermore this doesn't look very strong (ie the lay usage of significance) I would simply say it was decreased.

Response: Thanks for the reviewer's suggestions. We corrected these statements.

Reviewer #3 (Remarks to the Author):

The authors have thoroughly addressed all my comments with new, or improved, data, and consequently have strengthened the manuscript considerably. Congratulations on a very nice story.

Response: Thanks a lot.